# Bacterial–fungal interactions in the neonatal gut influence asthma outcomes later in life

Rozlyn CT Boutin[1,2]*, Charisse Petersen[2], Sarah E Woodward[1,2],
Antonio Serapio-Palacios[2], Tahereh Bozorgmehr[2], Rachelle Loo[1,2],
Alina Chalanuchpong[1,2], Mihai Cirstea[1,2], Bernard Lo[1], Kelsey E Huus[1,2],
Weronika Barcik[2], Meghan B Azad[3], Allan B Becker[3], Piush J Mandhane[4,5],
Theo J Moraes[6], Malcolm R Sears[7], Padmaja Subbarao[6,8], Kelly M McNagny[9,10],
Stuart E Turvey[1,11], B Brett Finlay[1,2,12]*

[1]Department of Microbiology and Immunology, University of British Columbia,
Vancouver, Canada; [2]Michael Smith Laboratories, University of British Columbia,
Vancouver, Canada; [3]Children's Hospital Research Institute of Manitoba and
Department of Pediatrics and Child Health, University of Manitoba, WinnipegMB,
Canada; [4]Department of Pediatrics, University of Alberta, Edmonton, Canada;
[5]School of Public Health, University of Alberta, Edmonton, Canada; [6]The Hospital
for Sick Children, Toronto, Canada; [7]Department of Medicine, McMaster University,
Hamilton, Canada; [8]Department of Pediatrics, University of Toronto, Toronto,
Canada; [9]Department of Biomedical Engineering, University of British Columbia,
Vancouver, Canada; [10]Department of Medical Genetics University of British
Columbia, Vancouver, Canada; [11]Department of Pediatrics, University of British
Columbia, Vancouver, Canada; [12]Department of Biochemistry and Molecular
Biology, University of British Columbia, Vancouver, Canada

*For correspondence:
rozlyn.boutin@msl.ubc.ca (RCTB);
bfinlay@msl.ubc.ca (BBF)

Competing interests: The
authors declare that no
competing interests exist.

Reviewing editor: Antonis
Rokas, Vanderbilt University,
United States

**Abstract** Bacterial members of the infant gut microbiota and bacterial-derived short-chain fatty acids (SCFAs) have been shown to be protective against childhood asthma, but a role for the fungal microbiota in asthma etiology remains poorly defined. We recently reported an association between overgrowth of the yeast *Pichia kudriavzevii* in the gut microbiota of Ecuadorian infants and increased asthma risk. In the present study, we replicated these findings in Canadian infants and investigated a causal association between early life gut fungal dysbiosis and later allergic airway disease (AAD). In a mouse model, we demonstrate that overgrowth of *P. kudriavzevii* within the neonatal gut exacerbates features of type-2 and -17 inflammation during AAD later in life. We further show that *P. kudriavzevii* growth and adherence to gut epithelial cells are altered by SCFAs. Collectively, our results underscore the potential for leveraging inter-kingdom interactions when designing putative microbiota-based asthma therapeutics.

Asthma is a chronic airways disease affecting over 300 million people worldwide and at least one-in-ten children in developed countries (*Asher and Pearce, 2014*; *Vos and Global Burden of Disease Study 2013 Collaborators, 2015*; *Organization, W, 2008*). The most common form of asthma is an allergic-type disease driven by excessive type-2 inflammation involving immunoglobulin (Ig)E antibodies, T helper (Th) type 2 cells, and lung eosinophilia (*Lambrecht and Hammad, 2015*). In severe cases, there can also be involvement of type-17 inflammation involving Th17 cells and interleukin

(IL)−17 (*Lambrecht and Hammad, 2015*; *Irvin et al., 2014*; *Wang et al., 2010*). Despite its high global prevalence, the etiology of allergic asthma remains incompletely understood, the disease has no cure, and patients with severe asthma often fail to achieve adequate disease control with standard treatments.

Recent findings of associations between suboptimal establishment (dysbiosis) of the bacterial communities within the infant gut microbiota and an increased risk of developing childhood asthma (*Arrieta et al., 2018*; *Fujimura et al., 2016*; *Stokholm et al., 2018*) have generated intense interest in exploiting the gut microbiota for therapeutic purposes. Although the exact signatures of dysbiosis are variable across studies, states of asthma- and allergy-associated dysbiosis have been consistently associated with reduced levels of fecal short-chain fatty acid (SCFA) bacterial fermentation products including acetate, butyrate, and propionate (*Arrieta et al., 2018*; *Roduit et al., 2019*; *Arrieta et al., 2018*). SCFAs are well recognized for their anti-inflammatory effects in the context of disease (*Roduit et al., 2019*; *Trompette et al., 2014*; *Thorburn et al., 2015*; *Arpaia et al., 2013*; *Smith et al., 2013*; *Machiels et al., 2014*), indicating that key functional consequences of asthma-associated bacterial dysbiosis may be conserved across studies and cohorts. To date, studies investigating the dysbiosis-asthma paradigm have predominantly focused on the bacterial signatures of dysbiosis and highlighted an asthma-protective role for certain gut bacteria (*Boutin et al., 2020*; *Arrieta et al., 2015*). In contrast, two recent studies in human birth cohorts have indicated that fungal dysbiosis, characterized by overgrowth of certain fungi, co-occurs with and is often much more conspicuous than asthma-associated bacterial dysbiosis (*Fujimura et al., 2016*; *Arrieta et al., 2018*). A causal role for fungal dysbiosis during infancy and later asthma outcomes has yet to be elucidated.

In Ecuadorian infants from the Ecuador Life (ECUAVIDA) study from a rural (non-industrialized) district of Quinindé, Esmeraldas Province (*Cooper et al., 2015*), we recently showed that fungal dysbiosis characterized by a general increase in total fungal load and increased abundance of the yeast *Pichia kudriavzevii* (also known as *Candida krusei*, *Issatchenkia orientalis*, and *Candida glycerinogenes* [*Douglass et al., 2018*]) within the gut at 3 months of age was associated with an increased risk of developing atopy and wheeze, a phenotype associated with an increased risk of asthma, at age 5 years (*Arrieta et al., 2018*). This yeast is commonly identified in human gut mycobiota studies (*Suhr and Hallen-Adams, 2015*) and has been found as a gut microbe in other birth cohorts (*Heisel, 2015*; *Ward et al., 2018*). Moreover, in a subset of 123 subjects from the CHILD Cohort Study, we found evidence suggesting that Canadian infants from an industrialized setting at high risk of asthma also demonstrate overgrowth of *P. kudriavzevii* in 3-month stool samples relative to healthy infants (*Figure 1*). Overgrowth of *P. kudriavzevii* in the gut in early life may therefore represent a relevant and widely applicable model of asthma-associated early life gut fungal dysbiosis.

While it has been demonstrated in adult mice that dysbiosis induced by treatment with antibiotic or antifungal agents exacerbates features of allergic airway disease (AAD) following antigen sensitization and challenge during the period of microbial disruption (*Skalski et al., 2018*; *Li et al., 2018*; *Shao et al., 2019*; *Wheeler, 2016*), neonatal life is a unique period of parallel development and maturation of both the microbiota and immune system. Fungal dysbiosis associated with asthma in neonatal and adult life may therefore differ mechanistically both in terms of the specific microbiota community compositions involved and their immediate and long-term immunological consequences (*Barcik et al., 2020*). Accordingly, using overgrowth of *P. kudriavzevii* as a model of fungal dysbiosis, we sought to determine whether fungal dysbiosis in the neonatal period influences asthma outcomes later in life, and to identify which aspects of asthmatic immunopathology are affected.

To establish a causal role for early life fungal dysbiosis in asthma etiology and validate previous findings in the ECUAVIDA cohort, we exposed specific-pathogen-free (SPF) mice to *P. kudriavzevii* during the neonatal period and then used the house dust mite (HDM) model of AAD to induce airway inflammation at 6 weeks of age (*Willart et al., 2012*; *Schuijs et al., 2015*; *Figure 2A,B*). Pups were exposed to either *P. kudriavzevii* suspended in phosphate buffered saline (PBS) or PBS alone by painting the abdomen and face of lactating dams with these respective solutions every second day for 2 weeks following birth. The presence of *P. kudriavzevii* in the guts of pups born to *P. kudriavzevii*-treated animals during the 2-week treatment period was confirmed by colony counts from plated colon tissues (*Figure 3—figure supplement 1a*). Relative to *P. kudriavzevii*-naïve (control) animals, animals exposed to *P. kudriavzevii* during the first 2 weeks of life demonstrated increased lung inflammation during AAD later in life, as evidenced by increases in lung

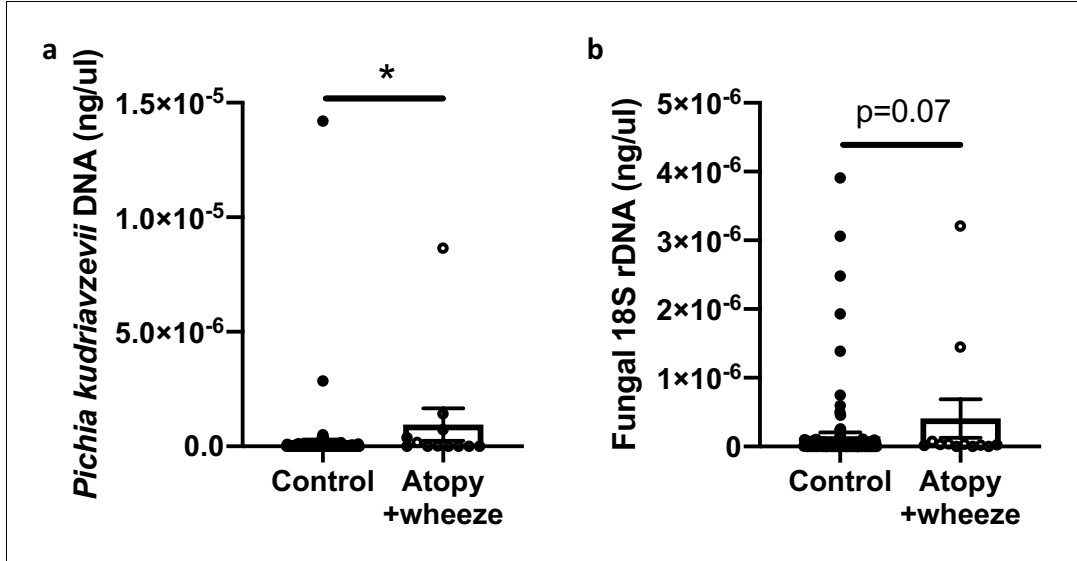

**Figure 1.** Canadian infants at high risk of asthma demonstrate increased levels of fecal fungi. Quantitative (q) PCR quantification (standard curve method) of DNA in feces collected at 3 months of age from infants in the CHILD Cohort Study (control subjects, n = 115; Atopy+Wheeze at age 5 years, n = 12). (a) qPCR quantification of *Pichia kudriavzevii* DNA. (b) qPCR quantification of all fungal 18S rRNA gene copies. Error bars represent the standard error of the mean and p-values were calculated using a Mann–Whitney test in GraphPad Prism; *p<0.05.

The online version of this article includes the following source data for figure 1:

**Source data 1.** qPCR quantification of *Pichia kudriavzevii* DNA in CHILD Cohort participants.
**Source data 2.** qPCR quantification of fungal DNA in CHILD Cohort participants.

histopathology scores, circulating IgE, lung eosinophils, and lung activated T cells expressing inducible co-stimulatory (ICOS) molecule (*Figure 2C–F*). ICOS expression is associated with IL-4 production (*Dong et al., 2001*) and Th2 responses (*Gonzalo et al., 2001*) in the lung and lymph nodes during asthma (*Uwadiae et al., 2019*). Moreover, ICOS is highly expressed on T follicular helper (Tfh) cells important for driving B cell class switching to IgE (*Beier et al., 2004*; *Reinhardt et al., 2009*; *Gong et al., 2019*), suggesting that *P. kudriavzevii*-exposed mice demonstrate increased adaptive immunity-driven lung inflammation. *P. kudriavzevii*-exposed animals also demonstrated an increase in the proportion and numbers of Th17 cells in the lung, identified by RORγt[high] expression and IL-17 secretion (*Figure 2G–H*), and greater numbers of lung GATA3+Th2 cells (*Figure 2I*). Notably, mice exposed to *P. kudriavzevii* for 2 weeks in adolescent life (4–6 weeks of age) via oral gavage did not show evidence of increased lung inflammation in the context of HDM-induced AAD (*Figure 2—figure supplement 1*), highlighting the importance of the previously reported 'critical window' of life during which the gut microbiota has the greatest ability to affect immune development relevant to asthma (*Arrieta et al., 2015*; *Stiemsma and Turvey, 2017*).

To further characterize fungal colonization in our model, we plated colon contents or fecal samples from pups born to *P. kudriavzevii*-treated dams immediately before and after weaning, when the gut microbiota is known to undergo dramatic shifts in community composition (*Al Nabhani et al., 2019*). Colony counts at days 16 (*Figure 3A*) and 21 (no colonies present) of life revealed that although levels were highly variable, *P. kudriavzevii* colonized the guts of pups born to dams treated with this yeast until at least 2 days after the final treatment, but was no longer present in the gut microbiota after pups were weaned on day 19 of life. Thus, pups were only colonized during the period when they were co-housed with dams and littermates, indicating that persistent exposure is required to maintain colonization (*Figure 3—figure supplement 1a*). This transient fungal colonization in early life has been previously described (*Fan et al., 2015*) and closely mimics the human condition wherein gut fungal populations decline over time in early life (*Schei et al., 2017*; *Kondori, 2019*) and may or may not stably colonize the adult gut (*Nash et al., 2017*; *Auchtung et al., 2018*) as a result of increasingly anaerobic conditions and colonization resistance,

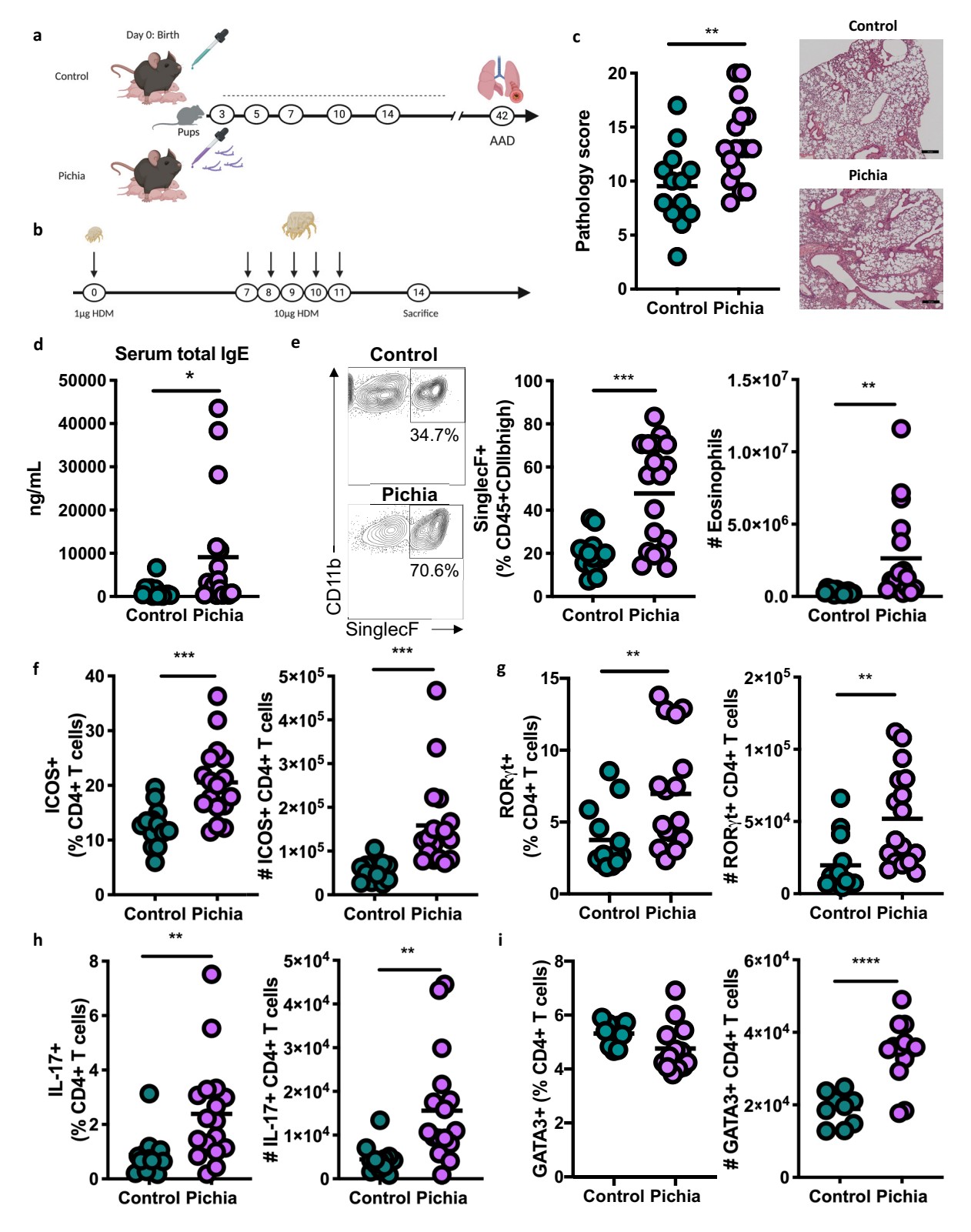

**Figure 2.** Mice neonatally exposed to *Pichia kudriavzevii* demonstrate increased inflammatory responses during allergic airway disease later in life. (a) Experimental procedure for neonatal exposure to *P. kudriavzevii* (Pichia; numbers indicate days of life) and (b) house dust mite (HDM) model of allergic airway disease (AAD). Dashed lines indicate the period of neonatal exposure in the pups. (c–i) Control and Pichia mice were born to dams treated with either PBS or yeast cells for 2 weeks after giving birth, respectively, and intranasally sensitized and challenged with HDM extract. (c) Lung pathology

*Figure 2 continued on next page*

*Figure 2 continued*

scores (left; 4–20) and representative images (right; 4× objective). (d) ELISA detection of serum IgE. (e) Representative eosinophil staining (left; pre-gated on single CD45+CD11 b$^{high}$ cells), frequency (middle), and total numbers of eosinophils (right) in the lung. (f) Frequencies (left) and numbers (right) of ICOS+ T cells (pre-gated on CD3+CD4+ cells) in the lung. (g) Frequencies (left) and numbers (right) of RORγt$^{high}$ T cells (pre-gated on CD3+CD4+ cells) in the lung. (h) Frequencies (left) and numbers (right) of IL-17+T cells (pre-gated on CD3+CD4+ cells) in the lung. (i) Frequencies (left) and numbers (right) of GATA3+T cells (pre-gated on CD3+CD4+ cells) in the lung. Data in (c–h) are pooled from three independent experiments each showing the same trends (control $n = 13$, Pichia $n = 17$). Data in (i) are pooled from two independent experiments each showing the same trends (control $n = 9$, Pichia $n = 13$). Dots represent individual mice and lines indicate the group mean. *p<0.05, **p<0.01, ***p<0.001; unpaired two-tailed Student's t-test with Welch's correction.

The online version of this article includes the following source data and figure supplement(s) for figure 2:

**Source data 1.** Neonatal exposure lung cell counts.
**Source data 2.** Neonatal exposure serum IgE.
**Source data 3.** Neonatal exposure lung histology scoring.
**Source data 4.** Neonatal exposure lung histology.
**Source data 5.** Neonatal exposure lung pathology.
**Figure supplement 1.** Adolescent exposure to *Pichia kudriavzevii* does not alter inflammatory responses during allergic airway disease later in life.
**Figure supplement 1—source data 1.** Adolescent exposure lung cell counts.
**Figure supplement 1—source data 2.** Adolescent exposure histology scoring.

among other factors. The absence of robust fungal communities in these animals at 4 and 8 weeks of age was verified by assessing for the presence of fungi in DNA isolated from fecal samples using high-throughput sequencing and primers targeting the internal transcribed spacer region (ITS-2) of the fungal 18S rRNA gene (sequencing files generated <1 Mb of raw data per sample).

Despite the absence of *P. kudriavzevii* in the gut past weaning, animals neonatally exposed to this yeast demonstrate increased levels of circulating *P. kudriavzevii*-specific IgG at 4 weeks of age and altered expression of a number of chymotrypsin-related genes previously associated with antigen-presenting immune cell function (*Naujokat et al., 2007*; *Chiba et al., 2014*), indicating that these animals mounted an immune response to this organism during the period of colonization (*Figure 3B*; *Figure 3—figure supplement 1b*). More specifically, RNA-sequencing analysis of colons from 16-day-old mice neonatally exposed to *P. kudriavzevii* demonstrated downregulated expression of *Try4*, *Cel*, *Cpa1*, *Cela3b*, *Cela2a*, *Prss2*, *Pnliprp1*, *Ctrl*, and *Ctrlb1* relative to *P. kudriavzevii*-naïve animals. Many of these genes have been implicated in immune functions related to bridging innate and adaptive immunity, including dectin-1-mediated signaling in dendritic cells (*Chiba et al., 2014*). Dectin-1 is a surface protein that plays an important role in sensing fungal β-glucan moieties (*Iliev et al., 2012*; *Goodridge et al., 2011*), suggesting a potential link between altered immunological function in the gut at the time of colonization with later HDM outcomes.

Non-bacterial microbes within the gut microbiota, even as transient colonizers, can dramatically and enduringly alter gut bacterial communities (*Martínez et al., 2018*; *Mason et al., 2012*; *Nieves-Ramirez, 2018*), and disruptions to gut bacterial populations have previously been linked to more severe asthma in the context of antibiotic-induced overgrowth of *Candida albicans* within the gastrointestinal tract (*Shao et al., 2019*; *Noverr et al., 2004*; *Noverr and Huffnagle, 2005*). Thus, we next examined whether early life exposure to *P. kudriavzevii* alters asthma-associated gut bacterial populations. Bacterial populations of *P. kudriavzevii*-exposed and -naïve mice separated according to treatment condition by principal component analysis based on Bray-Curtis Dissimilarity on day 16 of life (p=0.01; *Figure 3C*), and these differences persisted beyond the period of fungal colonization into adulthood (*Figure 3D–F*). Adult germ-free mice repopulated with colonic contents from 4-week-old SPF *P. kudriavzevii*-exposed animals (no longer containing *P. kudriavzevii* but with bacterial dysbiosis present; *Figure 3—figure supplement 2*, *Figure 3—figure supplement 3*), however, did not exhibit increased allergic airway inflammation following HDM sensitization and challenge relative to germ-free animals repopulated with colonic contents from *P. kudriavzevii*-naïve animals (*Figure 3—figure supplement 4*). These data indicate that actual presence of *P. kudriavzevii*, rather than differences in the gut bacterial populations at the time of AAD induction in *P. kudriavzevii*-exposed animals, is essential to the observed increase in severity of airway inflammation in conventional *P. kudriavzevii*-exposed animals.

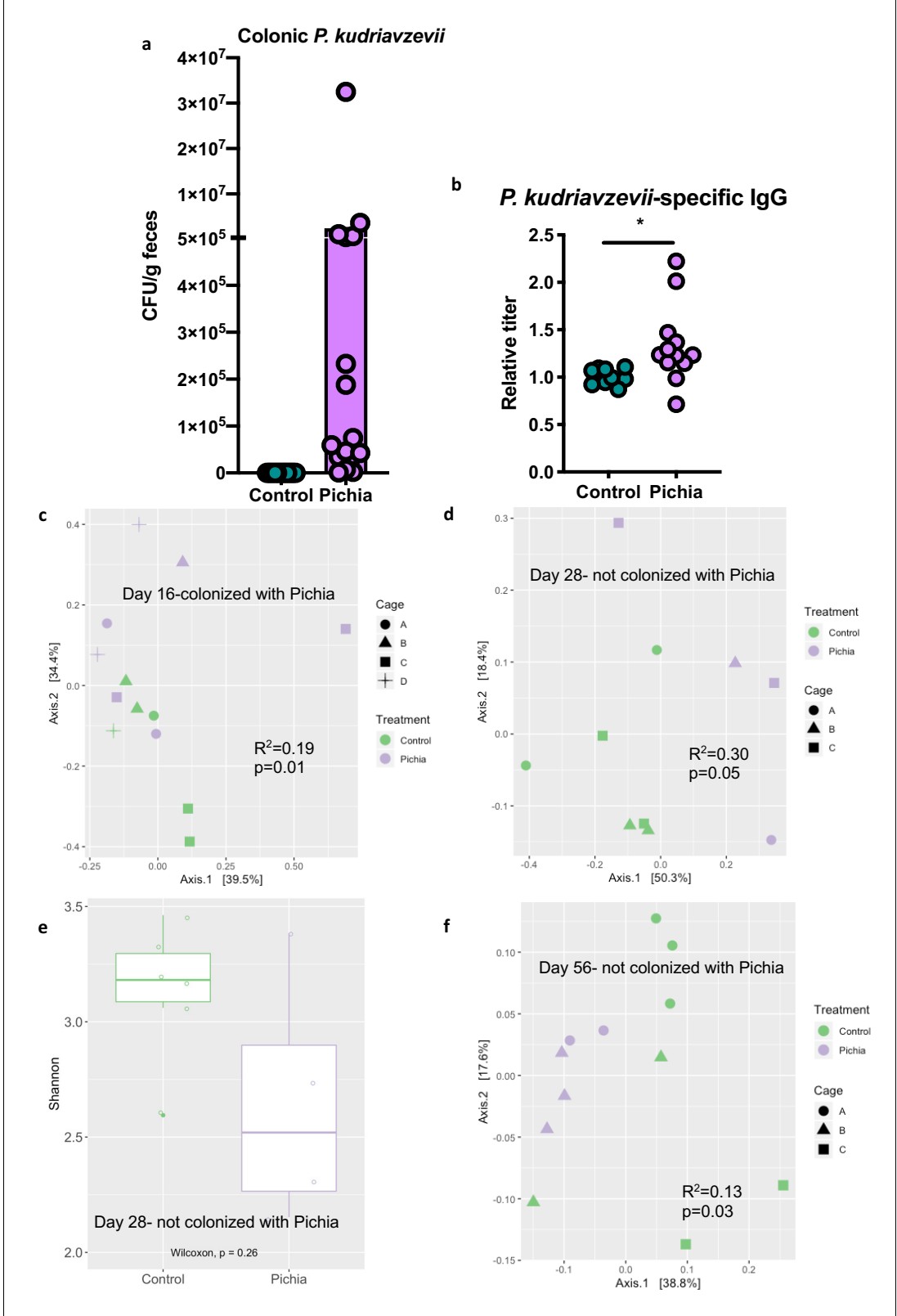

**Figure 3.** Mice transiently colonized with *Pichia kudriavzevii* in neonatal life mount an immune response to this yeast and exhibit persistent alterations to their gut bacterial populations. (a–c) Mice neonatally exposed to PBS (control) or *P. kudriavzevii* (Pichia) via suckling during the first 2 weeks of life were sacrificed at day 16 of life to assess for colonization by *P. kudriavzevii* or on day 28 of life for immunophenotyping. (a) Colony counts of *P. kudriavzevii* in colon tissues isolated from 16-day-old mice. (b) ELISA detection of serum *P. kudriavzevii*-specific IgG depicted as optical density (OD)
*Figure 3 continued on next page*

*Figure 3 continued*

absorbance value relative to controls sacrificed on day 28 of life. (c) Principal coordinate analysis (PCoA) plot based on Bray-Curtis Dissimilarity distances from 16S rRNA gene sequencing data from colonic contents collected at 16 days of age (control n = 6; Pichia n = 7). (d) PCoA plot based on Bray-Curtis Dissimilarity distances and (e) alpha diversity (Shannon index) derived from 16S rRNA gene amplicon sequencing data from fecal samples collected at 4 weeks of age (day 28) from mice treated as described in *Figure 2a* (control n = 6; Pichia n = 4). (f) PCoA plot based on Bray-Curtis Dissimilarity distances from 16S rRNA gene sequencing data from fecal samples collected at sacrifice (day 56) for animals treated as described in *Figure 2a* (control n = 7; Pichia n = 5). Mice in (f) were siblings of mice in (c–e). Data in (a) are pooled from three independent experiments each showing the same trends (control n = 15, Pichia n = 16). Data in (b) are pooled from two independent experiments each showing the same trends (control n = 9, Pichia n = 12). (c–f) Data are representative of at least two independent experiments each showing the same trends. Colors indicate treatment and shapes indicate different cages within each treatment condition. Fungal colonization status (presence/absence of Pichia in the gut) is indicated in each panel. (c, d, and f) p-values determined by PERMANOVA and corrected for cage effects. Dots represent individual mice. *p<0.05; unpaired two-tailed Student's t-test with Welch's correction.

The online version of this article includes the following source data and figure supplement(s) for figure 3:

**Source data 1.** Neonatal exposure colony counts.
**Source data 2.** Neonatal exposure Pichia-specific IgG.
**Source data 3.** Neonatal exposure colony counts.
**Source data 4.** Neonatal exposure colony counts.
**Figure supplement 1.** Mice neonatally exposed to *Pichia kudriavzevii* are transiently colonized and mount an immune response in the gut.
**Figure supplement 1—source data 1.** Pichia colonization time course.
**Figure supplement 2.** Gut bacterial populations from mice neonatally exposed to *Pichia kudriavzevii* in PBS or PBS alone are faithfully transplanted into germ-free mice.
**Figure supplement 3.** Relative abundance of the bacterial genera identified in fecal samples based on 16S rDNA amplicon sequencing from (a) 4-week-old mice (day 28) described in *Figure 2a* and (b) ex-germ-free mice given a fecal transplant as described in *Figure 3—figure supplement 1a* 2 weeks after the fecal transplant.
**Figure supplement 4.** Changes to gut bacteria resulting from neonatal colonization with *Pichia kudriavzevii* are not responsible for increased lung inflammation observed during allergic airway disease later in life.
**Figure supplement 4—source data 1.** Germ-free mice lung cell counts.
**Figure supplement 4—source data 2.** Germ-free mice serum IgE.

As neonatal exposure to *P. kudriavzevii* itself seems to exacerbate AAD later in life, we investigated how the infant gut niche influenced its tractability to fungal colonization. Little is known about the growth characteristics of *P. kudriavzevii*, but it has been observed to form pseudohyphae under certain conditions (*Kurtzman, 1998*; *Oberoi et al., 2012*). *C. albicans*, an opportunistic-pathogenic yeast previously demonstrated to be associated with exacerbated AAD in the context of bacterial dysbiosis (*Skalski et al., 2018*; *Kurtzman, 1998*), requires hyphal formation for efficient epithelial cell adhesion (*Dalle et al., 2010*; *Matsubara et al., 2016*), which is inhibited by the presence of bacterial derived SCFAs (*Noverr and Huffnagle, 2004*) and other organic acids (*Cottier et al., 2015*). Furthermore, a reduced abundance of SCFA-producing Clostridiales within the neonatal gut microbiota has been linked to impaired colonization resistance to pathogens (*Kim et al., 2017*), including fungi (*Fan et al., 2015*). Given that SCFAs have also been demonstrated to protect against asthma development and to be reduced in abundance in stool from infants at risk of asthma in Ecuador (in conjunction with fungal dysbiosis), the CHILD cohort, and other birth cohorts (*Roduit et al., 2019*; *Trompette et al., 2014*; *Thorburn et al., 2015*; *Arrieta et al., 2015*; *Cait et al., 2018*; *Arrieta et al., 2018*), we next investigated whether the asthma-protective effects of SCFAs could be mediated in part via their ability to prevent colonization of the infant gut by asthma-associated fungi.

We cultured *P. kudriavzevii* in the presence of physiologically relevant concentrations of acetate, butyrate, and propionate (*Arrieta et al., 2018*; *Arrieta et al., 2018*) at colonic pH 6.5 (*Nugent, 2001*) and observed that the presence of these molecules in the growth medium inhibits the growth of *P. kudriavzevii* (*Figure 4A–D*). SCFA-associated growth inhibition was accompanied by decreased pseudohyphae formation by *P. kudriavzevii*, as observed by scanning electron microscopy (*Figure 4E–I*). Pseudohyphae formation and its importance for gut colonization have not been previously reported in the context of commensal *P. kudriavzevii*, so we next interrogated whether this phenotype is required for colonization within the gut. *P. kudriavzevii* was pre-cultured in growth medium supplemented with acetate, butyrate, or propionate and its adherence to TC7 intestinal epithelial cells was measured. *P. kudriavzevii* cells grown in the presence of SCFAs demonstrate an

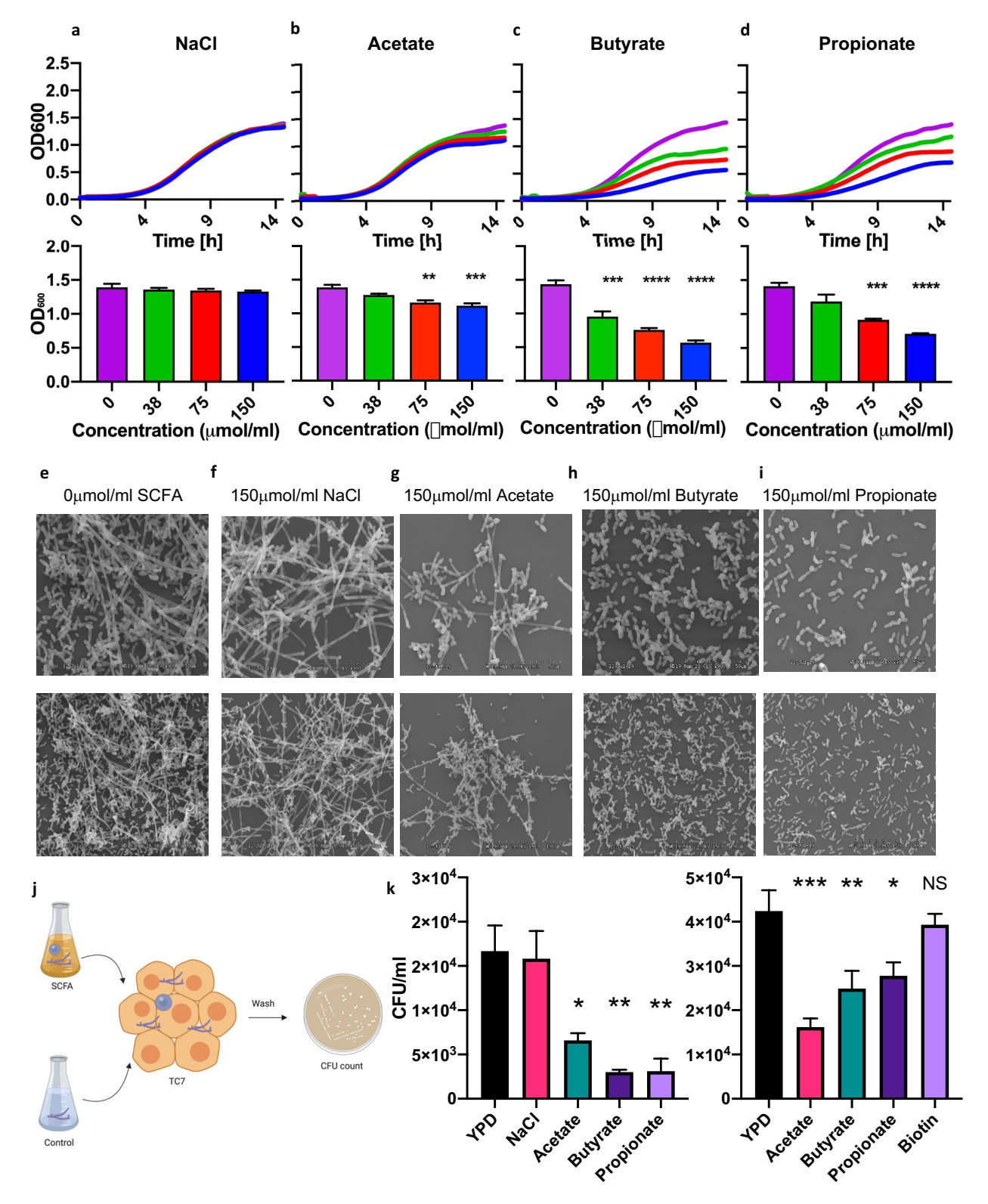

**Figure 4.** Short-chain fatty acids inhibit the growth of *Pichia kudriavzevii*. (**a–d**) Growth over time (top; mean of three biological replicates) and optical density (OD) at 600 nm at 15 hr (bottom) of *P. kudriavzevii* grown in yeast peptone dextrose (YPD) broth supplemented with sodium chloride (**a**) or the sodium salts of the short-chain fatty acids (SCFA) acetate (**b**), butyrate (**c**), or propionate (**d**) at the indicated biologically relevant concentrations (shown in units equivalent to μmol/g stool). (**e–i**) Scanning electron microscopy of *P. kudriavzevii* grown in YPD (**e**) or 150 μmol/mL of sodium chloride (NaCl) (**f**)

*Figure 4 continued on next page*

*Figure 4 continued*

or the sodium salts of acetate (**g**), butyrate (**h**), or propionate (**i**) at high (top) and low (bottom) magnification. (**j**) Epithelial cell adhesion assay experimental setup for (**k**). (**k**) Colony counts (colony forming units; CFU) of *P. kudriavzevii* pre-cultured in the presence of solutions described in (**e–i**) adherent to TC7 cells after 2 hr (left) and of *P. kudriavzevii* pre-cultured in the presence of the short-chain fatty acids (SCFA) acetate (150 μmol/mL), butyrate (30 μmol/mL), or propionate (30 μmol/mL) at biologically relevant molar ratios (shown in units equivalent to μmol/g stool) or biotin (10 mg/L) adherent to TC7 cells after 2 hr (right). (**a–d and k**) Data represent results from four independent experiments performed in triplicate. Dots represent biological replicates and unless otherwise stated, data are presented as mean ± SEM. Statistical comparisons are relative to SCFA-free controls. *p<0.05, **p<0.01, ***p<0.001, NS: not significant; ANOVA with Tukey's post hoc test.

The online version of this article includes the following source data and figure supplement(s) for figure 4:

**Source data 1.** SCFA growth curve.
**Source data 2.** SCFA growth curve.
**Source data 3.** SCFA growth curve.
**Source data 4.** SCFA growth curve.
**Source data 5.** SCFA growth curve.
**Source data 6.** SCFA growth curve.
**Source data 7.** SCFA growth curve.
**Source data 8.** SCFA growth curve.
**Source data 9.** SCFA growth curve.
**Figure supplement 1.** Mice supplemented with short-chain fatty acids (SCFAs) exhibit reduced colonization by *Pichia kudriavzevii*.
**Figure supplement 1—source data 1.** SCFA supplementation colonization.

impaired ability to adhere to these cells, with propionate and butyrate again having the most potent effects (*Figure 4K*). Furthermore, mice supplemented with a cocktail of SCFAs in their drinking water exhibited a trend toward reduced colonization with *P. kudriavzevii* following antibiotic treatment and fungal oral gavage (*Figure 4—figure supplement 1*). These findings indicate that commensal bacterial-derived SCFA production in early life may, in addition to having previously documented direct beneficial immunomodulatory and asthma-protective effects (reviewed in *Dang and Marsland, 2019*), prevent colonization of the infant gut by commensal fungi that have harmful asthma-associated immunomodulatory properties.

Herein, we have demonstrated a causal relationship between overgrowth of the commensal yeast *P. kudriavzevii* in the neonatal gastrointestinal tract and an increase in inflammation during AAD induced later in life, with involvement of both type-2 and type-17 inflammatory pathways. *C. albicans*, as well as *P. kudriavzevii*, are frequently found within the gastrointestinal tract but are also opportunistic pathogens blurring the line between commensal and pathogen (*Douglass et al., 2018*; *Shao et al., 2019*). Thus, host immune responses elicited by these microbes, particularly in early life and in the context of a disrupted gut bacterial microbiota, may have long-term consequences for immune-related diseases later in life. The immunological consequences of overgrowth of *P. kudriavzevii* in the neonatal gut are likely not exclusive to this organism. Overgrowth of different fungi, however, may result in subtle differences in the associated ecological and immune effects observed in the host. Mechanistically, early life fungal dysbiosis in the gut may drive long-lasting immune changes in the host through metabolite- or surface antigen-mediated (*Quintin et al., 2012*; *Netea et al., 2011*) influences on the early development of lymphoid organs (*Zhang et al., 2016*), innate, and/or adaptive immune cells. For instance, early exposures to fungi may modulate asthma-relevant immunity through the generation of memory immune cells in the gut that cross-react with common allergens such as HDM encountered in the lung. A similar phenomenon has recently been reported in human adults, wherein *C. albicans*-specific Th17 cells generated in the gut cross-react with airborne *Aspergillus fumigatus* to contribute to pathological airway inflammation during acute allergic bronchopulmonary aspergillosis (*Bacher et al., 2019*).

We also show that SCFAs, bacterial metabolites with direct immunomodulatory effects on the host, can inhibit the growth and morphology of asthma-associated fungi in a manner that has important functional consequences for gut colonization and subsequent immunomodulation. Taken together, our results suggest that gut bacterial communities with a reduced capacity for SCFA production create conditions permissive to invasion by transient fungal colonizers. In the neonatal gut, transient fungal colonization, in turn, may either directly or indirectly, through disruption to the normal temporal succession of neonatal gut microbiota communities required for appropriate immune

development, further alter immune development and susceptibility to asthma (*Figure 5*). These data highlight the importance of inter-kingdom interactions in determining microbiota-associated asthma outcomes and reveal a novel role for bacterial-derived SCFAs in protecting against asthma.

## Materials and methods

### Quantitative PCR

DNA was isolated from 123 stool samples collected at 3 months of age from a subset of subjects in the CHILD Cohort Study and selected based on stool and 16S rRNA gene sequencing data availability from the same sample or subject (samples for 16S rDNA sequencing were selected as previously described [*Boutin et al., 2020*]). All samples had a minimum DNA yield of 8 ng/µL following the DNA extraction. DNA was extracted from frozen stool samples using the Qiagen QIAmp PowerFecal DNA extraction kit according to the manufacturer's instructions. Total fungal load was assessed using the FungiQuant quantitative PCR (qPCR) assay on all samples submitted for ITS-2 rDNA gene sequencing (*Liu et al., 2012*). Specifically, sample DNA concentrations were determined by Qubit analysis and concentrations were normalized to 1 ng/µL, 10 ng/µL, or 100 ng/µL. 2 µL of template DNA was added to a reaction mixture containing iTaq Universal Probes Supermix (BIO-RAD), H2O, FAM probe (1 µM; Applied Biosystems), forward primer (10 µM; GGRAAACTCACCAGGTCCAG), and reverse primer (10 µM; GSWCTATCCCCAKCACGA) for a total reaction volume of 10 µL. This

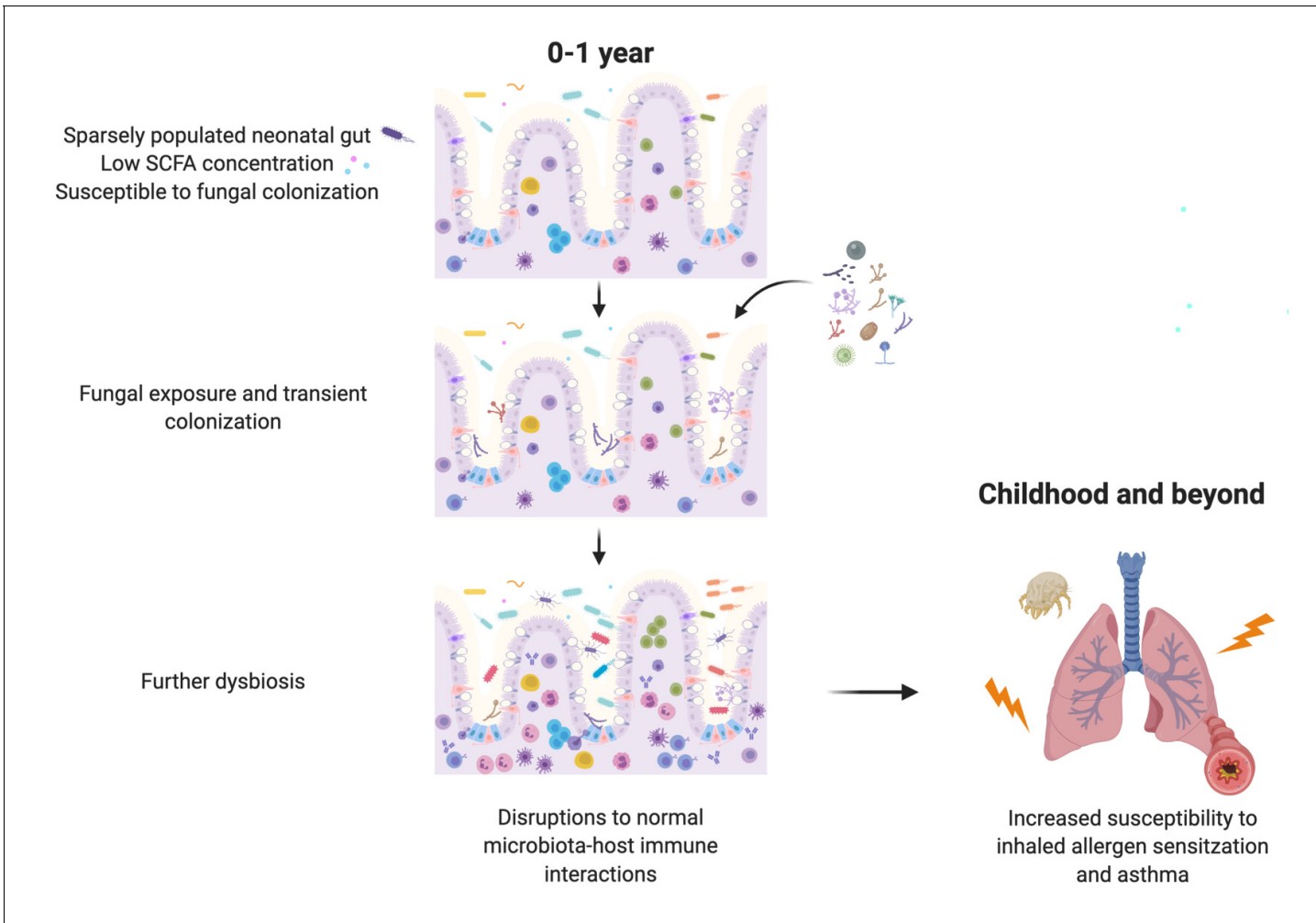

**Figure 5.** Schematic summary of the hypothesized sequence of events by which early life gut fungal dysbiosis associated with increased susceptibility to asthma in childhood occurs.

assay uses primers specific for the more highly conserved 18S rRNA gene of the fungal genome, which exhibits less length variability than the ITS-2 region and is therefore more suitable for qPCR assays. Reactions were run in duplicate at standard ramp speed, and qPCR was performed using the following cycling protocol: an initial enzyme activation step at 95℃ for 2 min followed by 45 cycles of a denaturation (95℃ for 15 s) step and then a combined annealing/extension step (60℃ for 1 min). Amplicon DNA concentration was determined using a standard curve generated using 10-fold dilutions of a 0.1 ng/μL stock of 18S rRNA gene amplicons purified from a PCR reaction completed using the FungiQuant primers and purified *Candida parapsilosis* template DNA. *C. parapsilosis* template DNA was extracted from a pure culture of *C. parapsilosis* (ATCC 22019) grown at 30℃ for 24 hr using the Quick-DNA Fungal/Bacterial Microprep Kit (Zymo Research) kit.

Total *P. kudriavzevii* load in CHILD samples and DNA isolated from mouse fecal samples or colon contents was assessed using previously validated qPCR primers specific for this yeast (*Heisel, 2015*; *Carvalho et al., 2007*). 2 μL of template DNA described above was added to a reaction mixture containing QuantiNova SYBR Green master mix (Qiagen), Rox reference dye (Qiagen), $H_2O$, forward primer (*Heisel, 2015*) (10 μM; CTGGCCGAGCGAACTAGACT), and reverse primer (*Carvalho et al., 2007*) (10 μM; TTCTTTTCCTCCGCTTATTGA) for a total reaction volume of 10 μL. This primer combination was selected based on its high efficiency, as calculated using a standard curve of *P. kudriavzevii* DNA, and ability to specifically target *P. kudriavzevii* at cycle threshold values below 30. This assay uses primers which target a *P. kudriavzevii*-unique sequence within the ITS-2 region. Reactions were run in duplicate and qPCR using the following cycling protocol: activation at 95℃ for 2 min followed by 40 cycles of denaturation (95℃ for 5 s) and annealing/extension (60℃ for 30 s). Cycling was performed at max/fast ramp rate. All qPCR reactions were performed on a 7500 Fast Real-Time System (Applied Biosystems, Foster City, Calif) machine and three negative controls were included on each plate. For the CHILD study samples, *P. kudriavzevii* DNA concentration was determined using a standard curve generated using 10-fold dilutions of a 1 ng/μL stock of 18S amplicons purified from a PCR reaction completed using the mentioned primers and purified *P. kudriavzevii*. *P. kudriavzevii* ( ATCC 6258) was grown at 30℃ for 24 hr using the Quick-DNA Fungal/Bacterial Microprep Kit (Zymo Research) kit.

Any samples with a large discrepancy between duplicate readings were re-run in triplicates. All samples were standardized to 10 ng/μL of total DNA for final comparative analysis. If DNA was not detected or the threshold value (Ct) was above the negative controls, the sample was considered to have no fungal DNA present. Any quantity of DNA detected in the negative controls was subtracted from all samples of the corresponding plate.

## CHILD Cohort Study case definition

Based on previously described definitions of cases and controls (*Arrieta et al., 2018*), subjects with 'Atopy+wheeze' were defined as those who wheezed in the past 12 months and had a positive skin prick test (sensitization/atopy) at age 5. Controls are those with no wheeze ever and no atopy at age 5 years. IgE-mediated allergic sensitization (atopy) was diagnosed based on skin prick testing to multiple common food and environmental inhalant allergens, using ≥2 mm average wheal size as indicating a positive test relative to the negative control (glycerin). Wheeze was assessed at age 5 years according to the Child Health Questionnaire or clinical assessment (*Subbarao et al., 2015*).

The CHILD Cohort Study protocols were approved by the human clinical research ethics boards at all universities and institutions directly involved with the CHILD cohort (McMaster University, University of British Columbia, the Hospital for Sick Children, University of Manitoba, and University of Alberta).

## Mice

C57BL/6J SPF mice (Jackson Laboratories, Bar Harbor, ME) were maintained in the Modified Barrier Facility at the University of British Columbia on a 12 hr light/dark cycle and provided with food and water ad libitum. Dams were maintained on a breeder diet and pups received normal chow after weaning. Germ-free mice were maintained in the Center for Disease Modeling at the University of British Columbia and housed in a cage-in-a-cage isolator system for the duration of the experiment. All experiments were in accordance with the University of British Columbia Animal Care Committee guidelines and approved by the UBC Animal Care Committee.

### Neonatal exposure to *P. kudriavzevii* or PBS

Glycerol stocks of $2 \times 10^7$ *P. kudriavzevii* (ATCC 6258) cells were prepared from 48 hr cultures generated from a single colony inoculated into YPD broth shaking at 37°C. Immediately prior to use, cells were spun at $6000 \times g$ for 5 min at 4°C and resuspended in 250 μL PBS (GE Healthcare Life Sciences). On days 3, 5, 7, 10, and 14 after giving birth, dams bred in-house or ordered directly from Jackson laboratories at day E15–17 of gestation were treated with *P. kudriavzevii* in PBS or PBS alone by scruffing the animal and using a P1000 pipette to gently apply 250 μL of solution to the abdomen and face region. Once pups reached 6 weeks of age, AAD was induced in these animals.

### Adolescent exposure to *P. kudriavzevii* or PBS

Four-week-old mice were given an oral gavage with $10^6$ cells of *P. kudriavzevii* suspended in PBS or PBS alone 30 min following an oral gavage with a 5% sodium bicarbonate solution on days 0, 2, 4, 7, 9, and 11. On day 15, AAD was induced in these animals.

### HDM-induced AAD

AAD was induced in 6-week-old animals as previously described (*Willart et al., 2012*; *Schuijs et al., 2015*). Briefly, animals were anesthetized with 3% isoflurane and sensitized with 1 μg of HDM protein (Greer Laboratories) in 40 μL of PBS by the intranasal route using a P200 pipette on day 0. Mice were subsequently anesthetized and intranasally challenged on 5 consecutive days 1 week later (days 7–11 of the model, inclusive) with 10 μg HDM protein in 40 μL of PBS. Three days after the final challenge (day 14 of the model), animals are sacrificed by intraperitoneal injection of 500 mg/kg tribromoethanol (Avertin; Sigma) and cervical dislocation.

### Isolation of lung immune cells

Lung tissues were excised and placed in complete RPMI (Gibco) supplemented with 10% fetal bovine serum (Gibco), 50 U/mL penicillin/streptomycin (Gibco), 1 mM sodium pyruvate (Gibco), 1× Minimum Essential Medium nonessential amino acids (Gibco), and 1× Glutamax (Gibco) until processed. Tissues were then cut into pieces using razor blades and shaken at 37°C for 45 min in 1× PBS with calcium and magnesium (GE Healthcare Life Sciences or Gibco) containing 5% (v/v) fetal bovine serum (Gibco) and 0.5 mg/mL Collagenase (Sigma catalog #C2139). Tissues were vortexed every 15 min during this tissue digestion step. Digested cells were passed through a pre-wetted 70 μM filter into ice-cold 1× PBS. Cells were collected by centrifugation at 4°C for 10 min at $800 \times g$ and then red blood cells were lysed using ammonium–chloride–potassium lysing buffer (Gibco). A hemocytometer and trypan blue staining were then used to manually count the isolated cells. Cells were normalized to an equal quantity among all samples prior to staining for flow cytometry.

### Flow cytometry and antibodies

Surface staining for flow cytometry was done in column buffer (2 mM EDTA [Millipore], 10 mM HEPES [GE Healthcare Life Sciences Life Sciences], 5% [v/v] fetal bovine serum in PBS without calcium and magnesium) for 20 min at 4°C. After staining, cells were washed twice with column buffer, fixed overnight in a 1:1 fix solution of column buffer: 4% paraformaldehyde, and washed prior to running flow cytometry. For intracellular cytokine staining, cells were stimulated for 8–12 hr with a Cell Stimulation Cocktail plus protein transport inhibitors (eBiosciences) at 4°C. Intracellular staining was done after fixing and permeabilizing cells using Perm/Fix buffer (eBiosciences) overnight at 4°C or for 1 hr at room temperature. Cells were then washed and stained in a Perm/Buffer wash solution (eBiosciences) for 30 min at 4°C. Cells were then washed twice with the Perm/Buffer wash before being resuspended in column buffer. Flow cytometry data for conventional mouse experiments were collected with a BD LSRII-561 and analyzed with FlowJo (Version 10) software. Flow cytometry data for the germ-free experiment was collected on an Invitrogen Attune NxT or CytoFLEX (Beckman Coulter) machine. See *Table 1* for antibodies used in this study. Single stain and Fluorescence Minus One controls were used for gating.

### Lung histology

Lungs were collected and fixed in 10% formalin for 48–72 hr, washed with 70% ethanol, and cut longitudinally into 5 μm sections following paraffin embedding (Wax-It Histology Services, Vancouver,

**Table 1.** Anti-mouse flow cytometry antibodies used in this study.

| Cell marker | Fluorophore | Source | Clone |
|---|---|---|---|
| CD3[†] | eFluor450 | eBioscience | 17A2 |
| CD4 | PerCP-Cy5.5 | TONBO biosciences | RM4-5 |
| CD4 | FITC | TONBO biosciences | RM4-5 |
| FOXP3 | PE | eBioscience | FJK-16s |
| RORγt | APC | eBioscience | B2D |
| ICOS | FITC | eBioscience | 7E.17G9 |
| GATA3 | PE-Cy7 | eBioscience | TWAJ |
| CD11b[†] | eFluor450 | eBioscience | M1/70 |
| CD11b | APC | eBioscience | M1/70 |
| CD11c | eFluor450 | eBioscience | N418 |
| F4/80 | FITC | eBioscience | BM8 |
| CD45 | PerCPCy5.5 | eBiosceicne | 30-F11 |
| SinglecF | PE | BD Biosciences | E50-2440 |
| GR-1 | PE-Cy7 | eBioscience | RB6-8C5 |
| IL-13 | APC-eFluor780 | eBioscience | eBio13A |
| IL-5 | PE | eBioscience | TRFK5 |
| IFN-γ | AlexaFluor700 | eBioscience | XMG1.2 |
| IFN-γ | PE | TONBO biosciences | XMG1.2 |
| IL-4 | APC | eBioscience | 11B11 |
| IL-17A | PE-Cy7 | eBioscience | eBio17B7 |
| B220[†] | eFluor450 | eBioscience | RA3-6B2 |
| NK1.1[†] | eFluor450 | eBioscience | PK136 |
| FcεRI[†] | eFluor450 | eBioscience | MAR-1 |

[*]Cy = cyanine.

[†]Used to define lineage + cells.

Canada). Sections were stained with hematoxylin and eosin and then blindly assessed for signs of inflammation. Histology was assessed as previously described (*Russell et al., 2012*) with minor modifications. Briefly, using the 4× objective, a score of 1–5 (1 = no signs of disease; 5 = severe disease) was assigned to each section for each of the following parameters: (1) peribronchial infiltration, (2) perivascular infiltration, (3) parenchymal infiltration, and (4) epithelium damage for a maximum score of 20. The 10× objective was used as needed to assess finer details.

## Serum collection and antibody measurements

Blood was collected by cardiac puncture immediately after sacrifice, allowed to clot, and serum was collected and stored at −70℃ until use. IgE levels were assessed by ELISA (ThermoFisher Scientific) according to the manufacturer's instructions. Serum levels of *P. kudriavzevii*-specific IgG were assessed by ELISA based on previously described methods (*Zeng et al., 2016*). High-binding plates were coated overnight at 4℃ with *P. kudriavzevii* from a 24 hr culture heat-killed at 90℃ for 1 hr and normalized to an $OD_{600}$ of 0.5 in 0.1 M sodium carbonate (Fisher Scientific) (pH 9.5). Plates were then washed four times with 0.05% Tween 20 (Sigma) in PBS and blocked for 2 hr at 37℃ with 2% bovine serum albumin (Sigma) in PBS. Samples diluted in PBS with 10% fetal bovine serum were then added and plates were incubated at room temperature for 2 hr. Plates were washed four times again and goat anti-mouse IgG antibody (Invitrogen; catalog #62–6540) diluted 1:1000 in PBS with 10% fetal bovine serum was added for 1 hr at room temperature. After four more washes, plates were incubated with Streptavadin-HRP (BD Biosciences, catalog #554066) diluted 1:1000 in PBS with 10% fetal bovine serum for 1 hr at room temperature. Plates were washed four times and TMB substrate (BD Biosciences) was added for 15 min. Reactions were then quenched with 2 M hydrochloric

acid and absorbance was measured at 450 nm. Total IgE levels were determined using a standard curve according to the manufacturer's instructions and all other serum antibody levels are reported as absorbance readings normalized to controls. All samples were run in duplicate for each ELISA.

## Fungal colonization assessment

Fecal pellets, colonic contents, or whole tissue and contents were collected into 1 mL of PBS and homogenized with a tungsten bead in a FastPrep-24 instrument (MP Biomedical) for 1 min at speed 5.5 one to two times as needed. Homogenates were plated on Sabouraud Dextrose Agar supplemented with 50 mg/L chloramphenicol and 5 mg/L gentamycin (SDA + CG). Plates were incubated at 37°C and colonies were counted the following day. Colony counts were normalized to sample weight.

## Isolation and RNA-sequencing of gut immune cells

Sixteen-day-old mice neonatally treated as described above with either *P. kudriavzevii* suspended in PBS or PBS alone were sacrificed using isoflurane and $CO_2$. Colons were then resected, placed into RNAlater (Qiagen) after fecal material was gently removed mechanically, and stored at $-70°C$ until use.

Colonic RNA was extracted using the ThermoFisher GeneJet RNA kit according to the manufacturer's instructions and sent to GENEWIZ (South Plainfield, NJ, USA) for quality control, DNAse treatment, and RNA sequencing. Briefly, extracted RNA samples were treated with TURBO DNase (Thermo Fisher Scientific, Waltham, MA, USA) to remove DNA following manufacturer's protocol. The RNA samples were then quantified using Qubit 2.0 Fluorometer (Life Technologies, Carlsbad, CA, USA) and RNA integrity was checked using Agilent TapeStation 4200 (Agilent Technologies, Palo Alto, CA, USA).

RNA sequencing libraries were prepared using the NEBNext Ultra RNA Library Prep Kit for Illumina following manufacturer's instructions (NEB, Ipswich, MA, USA). Briefly, mRNAs were first enriched with Oligo(dT) beads. Enriched mRNAs were fragmented for 15 min at 94°C. First strand and second strand cDNAs were subsequently synthesized. cDNA fragments were end repaired and adenylated at 3' ends, and universal adapters were ligated to cDNA fragments, followed by index addition and library enrichment by limited-cycle PCR. The sequencing libraries were validated on the Agilent TapeStation (Agilent Technologies, Palo Alto, CA, USA) and quantified by using Qubit 2.0 Fluorometer (Invitrogen, Carlsbad, CA) as well as by quantitative PCR (KAPA Biosystems, Wilmington, MA, USA).

The sequencing libraries were pooled and clustered on two lanes of a HiSeq (4000 or equivalen) flowcell. After clustering, the flowcell was loaded on the instrument according to manufacturer's instructions. The samples were sequenced using a 2 × 150 bp Paired End (PE) configuration. Image analysis and base calling were conducted by the HiSeq Control Software (HCS). Raw sequence data (.bcl files) generated from Illumina HiSeq was converted into fastq files and de-multiplexed using Illumina's bcl2fastq 2.17 software. One mismatch was allowed for index sequence identification. Raw data is available in the NCBI sequence read archive (SRA) under Bioproject ID PRJNA706731 (http://www.ncbi.nlm.nih.gov/bioproject/706731).

## Fecal DNA isolation and 16S library preparation

Fresh fecal pellets were collected and immediately stored at $-70°C$ until use. DNA was extracted using the QIAmp PowerFecal DNA kit with minor modifications. After adding solution C1, samples were heated at 65°C, placed on ice for 5 min, and then bead beat twice for 1 min at speed 5.5 using a FastPrep-24 instrument (MP Biomedical). Following DNA isolation, DNA was quantified using a Nanodrop machine and stored at $-20°C$ until use. To prepare DNA for 16S sequencing, DNA was normalized to approximately 10–50 ng/μL and 2 μL of DNA was used in each PCR reaction.

16S PCR (10 μL 5× buffer, 1 μL MgCl, 1 μL forward primer, 1 μL reverse primer, 1 μL dNTPs, 33.5 μL water, 0.5 μL enzyme, and 2 μL template DNA per reaction) was done using Illumina-tagged and barcoded primers specific for the 16S V4 region (*Kozich et al., 2013*) and the following cycling protocol: 98°C for 2 min, followed by 25–30 cycles of 90°C for 20 s, 55°C for 15 s, and 72°C for 30 s, and then 72°C for 10 min. Reactions were run on a gel to ensure successful amplification and then samples for each library were gel extracted (Thermo Scientific GeneJET Gel Extraction Kit) and

quantified using a PicoGreen assay (Invitrogen). Sequencing was performed using v2 technology on an Illumina MiSeq with 30% PhiX. Raw sequencing data is deposited to the NCBI sequence read archive (SRA) under the Project 'Early life Pichia and asthma' (accession PRJNA624902; https://www. ncbi.nlm.nih.gov/sra/PRJNA624902).

## Germ-free fecal transplant

Fecal transplants into germ-free mice were performed as previously described (*Suez et al., 2014*) with minor modifications. Briefly, individual fecal samples were collected from six donor animals neonatally exposed as described above to either *P. kudriavzevii*-exposed or PBS alone on day 27 (week 4) of life, pooled, and homogenized in 2 mL of pre-reduced PBS containing 0.05% cysteine-HCl. Donor animals were born to two different dams in each treatment condition and housed in different cages after weaning. The fecal slurry was allowed to settle by gravity under anaerobic conditions for 10 min and recipient 12–16-week-old C57bl/6 germ-free female mice were given an oral gavage with 100 µL of the supernatant. Recipient mice were housed in two separate cages per treatment condition and HDM-induced AAD was induced 2 weeks following the fecal transplant. Successful transfer of colonic bacteria was verified via 16S sequencing and flow cytometry staining was performed as above with minor modifications.

## *P. kudriavzevii*-specific PCR

The absence of *P. kudriavzevii* DNA isolated from the fecal transplant inoculum used in germ-free experiments was verified on a 1% agarose gel following PCR using *P. kudriavzevii*-specific primers (*Carvalho et al., 2007*) and the TopTaq Supermix (Qiagen).

## SCFA growth inhibition assay

*P. kudriavevii* was grown at 37°C in YPD for 48 hr and then inoculated into YPD supplemented with the sodium salts of the SCFAs acetate (Fisher Scientific), butyrate (Sigma), or propionate (Sigma) at the indicated concentrations based on known concentrations of SCFAs in the gut (*Arrieta et al., 2015*; *Arrieta et al., 2018*) in a 96-well plate. Sodium chloride was used as an osmolarity control and biotin as a negative control where indicated and all solutions were normalized to pH 6.5. Cultures were normalized to an $OD_{600}$ of 0.05 and grown in triplicate for 24 hr in a SYNERGY H1 Microplate Reader (BioTek) at 37°C, with readings taken every 15 min. The 96-well plate was shaken for 10–20 s before each reading. $OD_{600}$ readings were corrected for medium blanks for each condition at each concentration and three technical replicates were included per growth condition.

## Scanning electron microscopy

Fungal cultures were grown in 96-well plates as described above for 24 hr, washed with phosphate buffer (0.1M, pH = 7.4), and then fixed in 4% formaldehyde with 2.5% glutaraldehyde in 0.05M sodium cacodylate (pH 6.5) at room temperature for 2 hr and then overnight at 4°C. The following day, samples were washed with sodium cacodylate (0.1M, pH = 7.4), and post-fixed with an osmium/tannic acid solution (1% $OsO_4$ and 0.1% tannic acid in 0.1M sodium cacodylate, pH = 7.4). Samples were then dehydrated through exposure to a graded ethanol series (10, 20, 30, 40, 50, 60, 70, 80, 90, 95, and 3 × 100%), critical point dried using a Tousimis CPD Autosamdri 815B, and mounted onto a 12.5 mm Al SEM pin stub using a microporous sample holder. Samples were sputter coated with 10 nM AuPd in a Cressington HR208 and images were acquired on a S2600 VP scanning electron microscope.

## Epithelial cell adhesion assay

*P. kudriavzevii* was grown at 37°C in YPD for 48 hr and then inoculated into YPD supplemented with the sodium salts of the SCFAs acetate, butyrate, or propionate at a concentration of 150 µmol/mL in a 96-well plate. Sodium chloride was used as an osmolarity control and all solutions were normalized to pH 6.5. All cultures were normalized to a starting $OD_{600}$ of 0.05 using a Teacan plate reader and left for 24 hr shaking at 37°C. Cultures were then normalized to an $OD_{600}$ of 0.02 in a 1:3 mixture of YPD: DMEM. TC7 cells were washed with PBS and 1 mL of diluted yeast cells from each condition were added to TC7 cells and left for 2 hr in a 37°C incubator. TC7 cells were then washed twice with PBS to remove unattached fungal cells and dissociated with Accutase (Invitrogen). Harvested TC7

cells (with *P. kudriavzevii* attached) were plated onto SDA + CG plates overnight at 37°C for colony counting. Three technical replicates in each condition were performed to obtain each biological replicate and the cultures used to inoculate the TC7 cells were also plated to ensure an equal number of cells was added per condition.

## SCFA supplementation in adult mice

Six- to seven-week-old male and female mice were housed two per cage and drinking water was supplemented with 0.5 mg/mL cefoperazone (Sigma catalog #62893-20-3) as previously described (*Noverr et al., 2005*) on days 0–3 to clear the intestinal bacterial microbiota. Half of the mice further had their water supplemented with a cocktail of SCFAs according to previously established protocols (*Smith et al., 2013*; *Cait et al., 2018*) for the duration of the experiment. The cocktail consisted of sodium acetate (67.5 mM), sodium propionate (25.9 mM), and sodium butyrate (40 mM), and the control animals received water that was pH and sodium matched (*Smith et al., 2013*). All water was filter sterilized and had Splenda added (8 g/L) to improve palatability. On day 3, all mice were given an oral gavage with $10^7$ cells of *P. kudriavzevii* obtained from a 48 hr culture generated from a single colony of yeast and grown at 37°C while shaking. Two days after the gavage, fecal samples were collected for plating. Uncolonized mice were removed from the analysis.

## Quantification and statistical analysis

### ASV construction and taxonomic assignment for 16S data

Demultiplexed forward and reverse 250 bp reads were merged, denoised, trimmed, and filtered for sequencing quality control using DADA2 (*Callahan et al., 2016*) in QIIME2 (*Bolyen et al., 2019*). Taxonomic assignment of the resulting amplicon sequence variants (ASVs) was performed using the Greengenes database (*DeSantis et al., 2006*) version 13-8-99-515-806. All subsequent filtering and analysis steps were done in R Studio using the phyloseq (*McMurdie and Holmes, 2013*) and vegan (*Philip, 2003*) packages. ASV tables were filtered to remove singleton ASVs and taxa present at least three times in at least 10% of samples. To account for differences in sequencing depth, ASV tables were rarefied prior to computing alpha and beta diversities.

### RNA sequencing analysis

RNA sequencing reads were aligned to the grcm38 mouse reference genome using 'hisat2' and then converted to raw feature counts using 'featureCounts' in Python3. The resulting feature count table was analyzed in R Studio (R version 4.0.3) for differentially abundant genes. Genes detected less than once per every million reads were removed from downstream analysis. Trimmed Mean of M-values (TMM) normalization was performed on the remaining genes using the package 'edgeR' (*Robinson et al., 2010*). Differentially abundant genes were identified by a likelihood ratio test (LRT) using 'edgeR'. To limit type 1 errors, multiple correction was applied using Benjamini–Hochberg procedure and only genes with a false discovery rate (FDR) cut off of 10% or less were considered significantly different.

### Statistical analysis

Statistical analysis was performed in GraphPad Prism (http://www.graphpad.com) or in RStudio (https://www.rstudio.com/). Student's t-test with Welch's correction was used for flow cytometry data and ANOVA was used when more than two groups were present, with post hoc Tukey's test for multiple comparisons. For microbiome data, differences in beta diversity between groups were computed using a permutational analysis of variance (PERMANOVA) by the *adonis* function in the package *vegan* (*Philip, 2003*). All experiments included at least two cages per treatment group so p-values are adjusted for cage effects by using the option by = 'margin'. Alpha diversities were compared using a Wilcoxon Rank-Sum test. Group comparisons for qPCR results were done in PRISM using the Mann–Whitney test. Unless otherwise stated, error bars represent the standard error of the mean of pooled data and statistical significance is represented by *$p<0.05$, **$p<0.01$, ***$p<0.001$, and ****$p<0.0001$. *Figure 2a,b*, *Figure 3—figure supplement 2a*, *Figure 4*, *Figure 4— figure supplement 1a* and *Figure 5* were created with Biorender.com. R code is included in the Source Data associated with this manuscript.

## Acknowledgements

We thank Ingrid Barta for preparing some tissue samples for histology, staff at Wax-It Histology Services, staff at UBC's Modified Barrier Facility and Center for Disease Modeling for animal care support, Andy Johnson for flow facility training and support, Derrick Horne for assistance with scanning electron microscopy, Dr. Hind Sbihi for assistance with cleaning CHILD Cohort Study phenotype data, and all of our colleagues in the Finlay laboratory. We are grateful to all the families who took part in the CHILD Cohort Study, and the entire CHILD team, which includes interviewers, nurses, computer and laboratory technicians, clerical workers, research scientists, volunteers, managers, and receptionists. **Funding**: This work was funded by grants from CIHR to BBF (PJT-148484; FDN-159935) and AllerGen (12CHILD). RCTB was supported by a Vanier Canada Graduate Scholarship, a University of British Columbia (UBC) Four Year Doctoral Fellowship, and a Vancouver Coastal Health-Canadian Institutes of Health Research (CIHR)-UBC MD/PhD Studentship Award. MBA holds a Canada Research Chair in the Developmental Origins of Chronic Disease and is a Fellow of the CIFAR Humans and the Microbiome Program. SET holds the Aubrey J Tingle Professorship in Pediatric Immunology and Canada Research Chair in Pediatric Precision Health. BBF is a UBC Peter Wall Distinguished Professor and CIFAR Senior Fellow of the CIFAR Humans and the Microbiome Program.

## Additional information

### Funding

| Funder | Grant reference number | Author |
| --- | --- | --- |
| Canadian Institutes of Health Research | Project Grant PJT-148484 | B Brett Finlay |
| Canadian Institutes of Health Research | Foundation Grant FDN-159935 | B Brett Finlay |
| AllerGen | 12CHILD | Meghan B Azad<br>Allan B Becker<br>Piush J Mandhane<br>Theo J Moraes<br>Malcolm R Sears<br>Padmaja Subbarao<br>Stuart E Turvey<br>B Brett Finlay |
| Canadian Institutes of Health Research | Doctoral: Vanier Canada Graduate Scholarships | Rozlyn CT Boutin |
| Vancouver Coastal Health AND CIHR | UBC MD/PhD Studentship Award | Rozlyn CT Boutin |

The funders had no role in study design, data collection and interpretation, or the decision to submit the work for publication.

### Author contributions

Rozlyn CT Boutin, Conceptualization, Data curation, Formal analysis, Supervision, Validation, Investigation, Visualization, Methodology, Writing - original draft, Project administration, Writing - review and editing; Charisse Petersen, Conceptualization, Data curation, Formal analysis, Investigation, Visualization, Writing - review and editing; Sarah E Woodward, Data curation, Formal analysis, Validation, Investigation, Methodology; Antonio Serapio-Palacios, Data curation, Validation, Investigation, Methodology; Tahereh Bozorgmehr, Data curation, Validation, Investigation; Rachelle Loo, Bernard Lo, Data curation, Investigation, Writing - review and editing; Alina Chalanuchpong, Data curation, Investigation; Mihai Cirstea, Conceptualization, Data curation, Formal analysis, Validation, Investigation, Writing - review and editing; Kelsey E Huus, Data curation, Investigation, Methodology, Writing - review and editing; Weronika Barcik, Investigation, Methodology, Writing - review and editing; Meghan B Azad, Resources, Data curation, Funding acquisition, Investigation; Allan B Becker, Piush J Mandhane, Theo J Moraes, Malcolm R Sears, Padmaja Subbarao, Resources, Funding acquisition, Investigation; Kelly M McNagny, Resources, Funding acquisition, Methodology, Writing - review and editing; Stuart E Turvey, Resources, Supervision, Funding acquisition, Investigation,

Project administration, Writing - review and editing; B Brett Finlay, Resources, Supervision, Funding acquisition, Project administration, Writing - review and editing

### Author ORCIDs
Rozlyn CT Boutin  https://orcid.org/0000-0003-1598-0104
Sarah E Woodward  http://orcid.org/0000-0002-6688-0595
Mihai Cirstea  http://orcid.org/0000-0003-4900-6385
Kelly M McNagny  http://orcid.org/0000-0003-4737-3499
B Brett Finlay  https://orcid.org/0000-0001-5303-6128

### Ethics
Human subjects: The CHILD Cohort Study protocols were approved by the human clinical research ethics boards at all universities and institutions directly involved with the CHILD cohort (McMaster University, University of British Columbia, the Hospital for Sick Children, University of Manitoba, and University of Alberta). Work in the Finlay/Turvey labs is conducted under the ethics certificate number H07-03120.

Animal experimentation: All animal experiments were in accordance with the University of British Columbia Animal Care Committee guidelines and approved by the UBC Animal Care Committee (protocols A17-0322 and A13-0344).

### Decision letter and Author response
Decision letter https://doi.org/10.7554/eLife.67740.sa1
Author response https://doi.org/10.7554/eLife.67740.sa2

## Additional files

### Supplementary files
- Source data 1. R code for RNA-seq data.

- Transparent reporting form

### Data availability
Data Availability: All data generated or analyzed during this study are included in the manuscript and supporting files. Bacterial sequencing data have been deposited in the NCBI SRA under accession code SUB7276684 (https://www.ncbi.nlm.nih.gov/sra/PRJNA624902). RNA-seq data is deposited under Bioproject ID PRJNA706731 (http://www.ncbi.nlm.nih.gov/bioproject/706731).

The following datasets were generated:

| Author(s) | Year | Dataset title | Dataset URL | Database and Identifier |
|---|---|---|---|---|
| Boutin RCT | 2021 | Early life Pichia and asthma: | https://www.ncbi.nlm.nih.gov/sra/PRJNA624902 | NCBI Sequence Read Archive, PRJNA624902 |
| Boutin RCT | 2021 | Pichia Exposure RNAseq | https://www.ncbi.nlm.nih.gov/bioproject/706731 | NCBI Sequence Read Archive, PRJNA706731 |

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
