## [Decision Letter]

**Acceptance summary:**

This manuscript describes the association between overgrowth of the yeast *Pichia kudriavzevii* in the gut of infant mice and humans and risk for asthma later in life. In addition to replicating their previous finding of this association in a different human population, this work shows that yeast overgrowth stems from interaction with gut bacteria, highlighting the importance of inter-kingdom interactions in understanding microbiota-associated asthma outcomes.

**Decision letter after peer review:**

[Editors’ note: the authors submitted for reconsideration following the decision after peer review. What follows is the decision letter after the first round of review.]

Thank you for submitting your work entitled "Bacterial-fungal interactions in the neonatal gut influence asthma outcomes later in life" for consideration by *eLife*. Your article has been reviewed by 3 peer reviewers, and the evaluation has been overseen by a Reviewing Editor and a Senior Editor. The reviewers have opted to remain anonymous.

Our decision has been reached after consultation between the reviewers. Based on these discussions and the individual reviews below, we regret to inform you that your work will not be considered further for publication in *eLife* at this time.

All of us thought that the premise of your study is extremely interesting. However, there were numerous concerns raised by the reviewers. The reviewers expressed a need for dramatic changes to improve the clarity of the presentation and its logic. Reviewers also expressed many concerns about the existence of claims not full supported by the data provided. All of these issues are explained in detail in the individual reviewers' comments below.

Should the authors think they can address these criticisms, *eLife* would be willing to reconsider a resubmitted manuscript.

Please note, however, two important points. First, please note that any resubmission would be considered as a new submission and may necessitate acquisition of comments from new reviewers (if one or more from the ones that reviewed this version are not available). Second, please note that your data and analyses will need to strongly support the conclusions (and the conclusions will need to be sufficiently exciting) in order for a re-submission to be successful.

*Reviewer #1:*

This manuscript describes the association between *Pichia kudriavzevii* overgrowth in the gut of infant mice and humans and the occurrence of allergic airway disease (AAD) later in life. Here, the authors replicated their previous finding in Ecuadorian children, which showed an association between fungal overgrowth, particularly levels of *P. kudriavzevii*, and AAD later in life, using a population of Canadian children. They went on to show a correlation in mice between early exposure to *P. kudriavzevii* while weaning and increased inflammation in an AAD model after fungal exposure ended. Building on their previous work which found lower levels of short-chain fatty acids in the population with higher *P. kudriavzevii*, which are produced by intestinal bacteria, they show that higher concentrations of butyrate and related molecules suppress growth, alter morphology and reduce adherence of *P. kudriavzevii* to Caco2/T7 epithelial cells leading them to propose a model that modulation of gut bacteria could be used to prevent the effects of fungal overgrowth and sensitization to allergens early in life. Overall, the paper is interesting, but the text is confusing with some errors and inconsistencies in places.

1. For all bar plots, please show individual data points (e.g. Figure 1) on the graphs to show the data distribution. This is necessary to confirm the authors conclusion that there is a correlation between individuals with AW have increased Pichia burden or if this is the case for only a subpopulation. The 18S-based fungal load data, in light of the y-axis, are not compelling-data points again should be shown. Why does there appear to be more Pichia DNA than total fungal DNA?

2. For experiments in which different data from each mouse is available, such as Pichia abundance and eosinophil counts, or different immunological markers, please perform a correlation analysis to determine if the variability seen between mice further supports the conclusions made in this paper instead of a series of independent t-tests. Do mice with the highest Pichia colonization have a stronger immune response? In multiple panels a few data points appear to drive the differences between the groups, if these data are from the same experiment in each panel that needs to be addressed in the text and in the statistical analysis. It would support the message of the paper.

3. What was the rationale for adding IL-4+ cell analysis in Figure 2-supplement 1F but not in Figure 2? The authors mention that ICOS is associated with IL-4 production, but IL-4 data are not shown. Furthermore, In Figure 3-supplemental 2 include IL-5 and IL-13 to show that the same response characterized in Figure 2, but these metrics were not present in Figure 2. Please either include the data or explain.

4. In multiple places in the text it is stated that Pichia is absent from the gut at different time points, however no data was shown to directly support this conclusion. Either add data or delete these statements.

a. Line 114 – no data shown for colony counts at 21 days, remove from text or add this missing figure.

b. Line 116 – no data is shown to support that Pichia is gone after weaning. Without showing that Pichia did not persist after weaning then, it is possible that prolonged fungal colonization is responsible for the AAD phenotype observed. Address this inconsistency by adding in the relevant data or addressing in the text.

c. Line 120-124 – no data shown to support that there is no Pichia in the gut at 4 weeks.

d. Line 131-132 – have not shown fungal colonization ends at a particular day.

5. A discussion of data in Figure 3C-F correlates with previously published studies on bacterial dysbiosis and AAD would be a beneficial addition. Clostridales are mentioned as a direct contributor to SCFA, is the abundance of members of this genus altered in the experiment shown?

6. In Figure 3-supplemental 1, data is shown to support the hypothesis that the bacterial composition change in the gut alone does not account for the changes in immune response however no data is shown to verify that the fecal transplant procedure successfully recapitulated the Pichia exposed gut microbiome. An analysis comparing data in Figure 3D and Figure 3 supplemental 1B-D is important to show that the bacterial species present in the fecal sample obtained from the mice exposed to Pichia were able to successfully colonize the germ-free mice and be representative of the original population. It would also be interesting to do the same analysis with the post-AAD samples to show how Pichia exposed and germ-free colonized mice (Figure 3F and Figure 3-supplemental 1E-F) respond to AAD model – is there a difference?

Changes to the text:

1. This study looks at the effects of Pichia on bacterial dysbiosis and immune response but does not show data to support a more general change in fungal dysbiosis (no other fungal species present in the mouse gut were assessed). Throughout the paper, as in line 92-95, this point should not be made without support.

2. The biological relevance of Figure 2 and Figure 2-supplement 1 are difficult to interpret without first having established colonization/dysbiosis of the gut microbiome. Consider moving Figure 3 first to establish you can recapitulate the dysbiosis seen in children in this mouse model and then show in Figure 2 that this correlates with differences in the immune system later in life.

3. The data in Figure 4 don't align with the other data in the paper without more discussion of the connection. Expand upon the discussion/relevance of assessing SCFA effects on Pichia in this paper. Have SCFA been shown to impact fungal colonization previously, in analyses that corelate SCFA with decreased AAD? Is the hypothesis being tested that before increased Pichia in children there was changes in the microbiome that decreased local SCFA concentrations and this allowed Pichia overgrowth, which in turn further impact the bacterial gut microbiome and immune development? Clarify a connection in the text as to why Figure 4 is important to the story arc of this paper. It might be useful to present both models in a concluding figure.

4. For Figure 3B, to address the relevance of circulating Pichia-specific IgG, compare data to fold changes in other studies in other systems.

5. Line 84 suggest that the data in Figure 1B are significant (p<0.05), but according to the figure they are not, edit text to acknowledge the p-value accordingly.

*Reviewer #2:*

In this report Boutin and colleagues have investigated the impact of exposing mice to *P. kudriavzevii* on HDM-induced allergic lung inflammation. The premise of the work comes from previous data from the CHILD study, amongst others, that link certain fungal species with an increased asthma severity.

There are two main conclusions from this work. First, the exposure must occur pre-weening and second, that SCFA directly reduce fungal adherence to intestinal epithelial cells.

Figure 1 is in line with prior reports, setting the rationale for the author's focus on Pichia. The best control to include would be a comparison with another fungi, which might serve as a negative control. As the authors indicate, there are already a number of reports showing anti-fungal treatment, or colonisation with specific fungi, exacerbate allergic inflammation in mouse models. Is there something unique to Pichia, or is a general outgrowth of any fungi seen as increasing susceptibility?

The new data starts with Figure 2 where in general there is an increase in adaptive immune parameters in the Pichia group. Variability is very high, and in many cases, there appear to be 'responders' and 'non-responders', e.g. panel's D, E, G, H. Could these be cage affects? Statistical significance is reached in most cases, likely due to the pooling of experiments to have sufficient numbers. In this respect, it is unclear why data is pooled from 2 experiments in some case, and 3 in others? What is the basis for this? Overall, while there are some differences, particularly in the FACS analysis, the overall phenomenon does not look particularly robust, and disease (as opposed to FACS) parameters are largely missing, e.g. mucus production and lung function. There is no correlation with the level of Pichia colonisation per mouse and the readouts e.g. did the mice showing high levels of eosinophils also have higher CFU of Pichia during the 2 weeks of exposure?

The concerns continue with Figure 2-sup where the authors have changed the protocol, now giving the Pichia per oral and post-weening. The conclusion is that the phenomenon no longer exists and thus the exposure must occur pre-weening. In fact, in this case there are far fewer animals and the figure legend indicates this is a single experiment (no repeats referred to?). The pathology score appears to have a similar increase in the Pichia group as in Figure 2, however an outlier in the control group likely stops significance. The fact that there is a lot of variability and so few mice, plus the fact that two different protocols have been used (skin administration vs oral gavage), means that the authors can't conclude the Pichia is only active in early life. These data do not appear robust and the conclusions are overstated.

In Figure 3 the highly variable exposure to Pichia is noted and the authors argue that there is an increase in Pichia specific IgG, however these data are shown as relative to control and there is a minor shift to around 1.2 in most samples. This analysis does not appear to be robust enough to conclude an adaptive response has been mounted against the Pichia. The clearest data from Figure 3 comes with the GF recolonisation experiment, which in fact shows that transferring the microbiome from Pichia mice does not transfer any susceptibility. The inferred conclusion therefore is that the effect must be linked with early life immune maturation. However, no evidence is provided to support this. Neonatal GF mice would need to be colonised as a control, and immune parameters measured to prove that there has been some early life imprinting. Moreover, given the need to pool multiple experiments to show their phenomenon in Figure 2, similar numbers of mice are likely required throughout the manuscript to allow robust conclusions.

Finally, in Figure 4 the authors show that culturing the fungi in vitro in the presence of SCFA reduces fungal adherence to an intestinal epithelial cell line. The data is clear, but the obvious concern is the physiological relevance of this, and also the absence of any link with the prior data in the manuscript. Ideally the authors would include a better negative control, showing that other common intestinal metabolites do not affect fungal adherence. In addition, the SCFA should be utilised at ratios similar to that seen in vivo. In Figure 4, the highest concentration of 150umol/ml is utilised for each SCFA, however if the butyrate and propionate levels were relative to acetate, the conclusion would likely change. Direct cell toxicity of the SCFA should also be reported. These data would also carry more weight if in vivo experiments were performed to test whether there is reduced Pichia colonisation in early life in mice with higher SCFA levels.

Overall, although conceptually the manuscript is interesting, there are major concerns related to the robustness of the datasets and comparisons that are drawn between the different experimental systems.

*Reviewer #3:*

Boutin et al. performed a throughout study to investigate the cause of allergic airway disease induced by early fungal colonization. They used the commensal fungus *Pichia kudriavzevii* as reasonable example in mouse models. They furthermore demonstrate that the disease is only induced when mice are exposed early in life and that the resulting disease is predominantly induced by the fungus and not the resulting microbiome.

I believe this is a well thought out and conducted study. The following points should be considered by the authors for improving the quality and clarity of their study:

The major problem in the manuscript is the unjustified experiments with SCFAs. Suddenly in the manuscript there is a statement that SCFAs have been reported in the literature to inhibit fungi so the authors decided to test some of them for their inhibitory role on P. kudriavzevii. Subsequently and after the successful testing of the inhibitory role of the tested SCFAs in vitro the authors claimed that they have found the mechanism behind colonization of P. kudriavzevii. The authors could support this in several ways:

– There is a huge variation in the *P. kudriavzevii* levels of the 115 subjects from the Child cohort. A 16S rRNA analysis of these subjects and a correlation analysis with *P. kudriavzevii* levels could prove that SCFA bacterial producers correlate negatively with *P. kudriavzevii* levels. The authors could measure the SCFAs in the stool samples of these children to confirm their hypothesis.

– In the mice experiments I would like to see that supplementation of the food with these SCFAs indeed leads to colonisation resistance against *P. kudriavzevii*.

– Alternatively, a concomitant oral gavage of a known strong SCFA bacterial producer and *P. kudriavzevii* could offer the evidence needed to support the authors hypothesis.

One of the most interesting findings was that: unlike the neonatal exposure, mice exposed to *P. kudriavzevii* for two weeks in adolescent life (4-6 weeks of age) via oral gavage did not demonstrate increased lung inflammation. However, it is not completely clear to me whether the two exposure groups ("neonatal exposure" and "adult exposure" as stated in the Methods) have the same treatment methods and dosage. Please elaborate more on this.

The authors stated in Line 387 of the Methods section that "Successful transfer of colonic bacteria was verified via 16S sequencing", but I did not find relevant results or figures.

Line 128: "Bacterial populations of P. kudriavzevii-exposed and -naïve mice separated moderately"

Please supply an effect size measure in the corresponding figure or text to elucidate what a 'moderate' effect is in this context.

Line 131: High variability and low sample size for a mice-based 16s rRNA analysis, in Figure 3C, 3D, 3F (as main results), although being statistically significant. The authors used Bray-Curtis Dissimilarity as community β-diversity measures, I wonder whether other distance measures (e.g. weighted/unweighted UniFrac) could have higher power in discriminating the two groups.

Line 132: I would suggest adding the points for individual samples in Figure 3E.

Line 272: Is four-week-old mice really "adult exposure" as stated in the subtitle? or, adolescent?

Line 383: were the recipient germ-free mice 12-16-week-old? Based on the figure legend of Figure 3-supplement 1a, AAD induction was at day 42 (week 6). Please confirm and/or clarify.

Figure 1: As statement by the authors, the variation is high with respect to fungal counts. If possible, scatter or bee swarm plots would help to assess the variation better.

Figure 2c In the caption, the pathology score is shown and referenced as "left; 5-20". However, controls clearly show a value below 5. Please correct the range or elucidate.

Figure 2-suppl c: Same as with Figure 2C, please correct the caption.

Figure 3c-d: Please report a proper effect size in the plot (such as F-values or R2) or in the caption in addition to the p-value.

Figure 3—figure supplement c "Relative abundances of the top 100 sequences at genus level […]"

"Top 100 sequences" does not make sense in this context. Please correct or elucidate. I guess it should be similar to Figure 3—figure supplement E.

Figure 3-suppl e "top 100 genera"

Only 19 genera are shown in the plot (which is reasonable). Please report it properly in the caption.

Please lower case the sub-figure letters in the figure captions (or vice versa)

---

## [Author Response]

[Editors’ note: the authors resubmitted a revised version of the paper for consideration. What follows is the authors’ response to the first round of review.]

Reviewer #1:This manuscript describes the association between Pichia kudriavzevii overgrowth in the gut of infant mice and humans and the occurrence of allergic airway disease (AAD) later in life. Here, the authors replicated their previous finding in Ecuadorian children, which showed an association between fungal overgrowth, particularly levels of P. kudriavzevii, and AAD later in life, using a population of Canadian children. They went on to show a correlation in mice between early exposure to P. kudriavzevii while weaning and increased inflammation in an AAD model after fungal exposure ended. Building on their previous work which found lower levels of short-chain fatty acids in the population with higher P. kudriavzevii, which are produced by intestinal bacteria, they show that higher concentrations of butyrate and related molecules suppress growth, alter morphology and reduce adherence of P. kudriavzevii to Caco2/T7 epithelial cells leading them to propose a model that modulation of gut bacteria could be used to prevent the effects of fungal overgrowth and sensitization to allergens early in life. Overall, the paper is interesting, but the text is confusing with some errors and inconsistencies in places.

Thank you for your feedback, we sincerely appreciate your detailed assessment of our work and appreciate the opportunity to address these important points.

1. For all bar plots, please show individual data points (e.g. Figure 1) on the graphs to show the data distribution. This is necessary to confirm the authors conclusion that there is a correlation between individuals with AW have increased Pichia burden or if this is the case for only a subpopulation. The 18S-based fungal load data, in light of the y-axis, are not compelling-data points again should be shown. Why does there appear to be more Pichia DNA than total fungal DNA?

Excellent point. As this reviewer rightly points out, this is a frequently observed finding in human microbiota studies and the increased Pichia burden is present only for a subpopulation of children in the CHILD cohort (6/12 cases vs 36/115 in the controls). This is also what we saw in our previous publication using data from a human birth cohort in Ecuador (Arrieta et al., (2018)) (1). We agree, however, that the data in the CHILD cohort are not as compelling as those in the original paper. The purpose of this data is to support our use of Pichia overgrowth as a relevant model, and our results suggest that larger sample sizes should be used in future work. Given that this is preliminary data, this has now been moved to be a Supplementary file and we have replaced the original figures with figures showing the individual data points. We have also qualified our wording and reduced the focus on this data:

Line 98-103: “Moreover, in a subset of 123 subjects from the CHILD Cohort Study, we found evidence suggesting that Canadian infants from an industrialized setting at high risk of asthma also demonstrate overgrowth of *P. kudriavzevii* in three-month stool samples relative to healthy infants (Supplementary file 1). Overgrowth of *P. kudriavzevii* in the gut in early life may therefore represent a relevant and widely applicable model of asthma-associated early life gut fungal dysbiosis.”

**Author response image 1. sa2fig1:** 

Thank you for pointing out the y-axes of our graphs, this review allowed us to look at the data more closely and further highlighted the utility of showing the individual data points as suggested. The y-axis of the graph for *P. kudriavzevii* DNA is brought up by an outlier in the control group, but with the individual data points shown it becomes more evident that the total fungal population in the samples is around 10^-6^ whereas the *P. kudriavzevii* load is less than this. The loads could appear similar for several reasons, and this again mimics what was seen in our 2018 JACI paper by Arrieta et al. Differences in the copy number of target DNA could account for these differences. Moreover, the standard curves for the Pichia figure used total Pichia DNA (full genome; ~4 ITS copies/genome), whereas those for the total fungal load analysis used only the 18S amplicons. Therefore, we would expect the Pichia figure to reflect more total DNA/18S copy. Finally, note that the assay used for the FungiQuant analysis (probe-based method) differs from the Pichia-specific (SYBR-based method) analysis and has a longer amplicon. If copy number is calculated (https://www.idtdna.com/pages/education/decoded/article/calculations-converting-from-nanograms-to-copy-number) after converting the Pichia DNA amount to ITS copies (based on the average genome size of 10.8126Mb(https://www.ncbi.nlm.nih.gov/genome/?term=Pichia%20kudriavzevii[Organism]&cmd=DetailsSearch) and 4 copies of the 18S rDNA locus per genome(2)), the results are shown in Figure 1.

2. For experiments in which different data from each mouse is available, such as Pichia abundance and eosinophil counts, or different immunological markers, please perform a correlation analysis to determine if the variability seen between mice further supports the conclusions made in this paper instead of a series of independent t-tests. Do mice with the highest Pichia colonization have a stronger immune response? In multiple panels a few data points appear to drive the differences between the groups, if these data are from the same experiment in each panel that needs to be addressed in the text and in the statistical analysis. It would support the message of the paper.

We agree with the reviewer that if this type of data were available, it would greatly strengthen our paper, and this has been the topic of much ongoing discussion within our group. Unfortunately, this correlation analysis is not possible due to the design of the experiment and technical challenges associated with obtaining colonization data for the specific individual pups used for the asthma experiments. We found that pups are only colonized with Pichia when they are housed with the dams (who are the ones that actually received the treatment), but lose colonization once they are weaned at 19 days of age. Therefore, it is not possible to regulate exactly how much Pichia each individual pup receives. Furthermore, before weaning, pups do not produce fully formed fecal samples so colonization has to be measured as an endpoint analysis-i.e. by collecting gut tissues. As a result, the colonization data is collected on different animals from those taken out to the AAD endpoint.

We also worry that colonization of pups would be variable within animals over the course of the first 2 weeks of life (given stochastic exposure, coprophagy, etc.) and are concerned that correlating Pichia burden at any given time point with inflammatory markers would be an inaccurate assessment.

Despite these limitations, it was important to us to address this concern. We therefore repeated the full 12-week experiment and fecal samples were collected during and after the colonization period to correlate Pichia load (determined using colony counts) with asthma outcomes. Unfortunately, this experiment confirmed that the earliest time point a formed fecal sample could be obtained from pups was 18 days of life and given the low colonization at this time point, no correlations were found between colony counts and asthma outcome parameters.

With regards to the “responder” vs “non-responder” mice, we did observe that female mice appear to be more sensitive to the effects of fungal dysbiosis on AAD (note that female and male mice were housed separately and both sexes were included in the study (1 cage/sex that received Pichia/experimental repeat except for 1 of the 3 repeats)). This is likely in part due to the increased sensitivity of female mice to animal models of allergic disease.

It’s important for us to mention that, while these figures reflect pooled data, our three independent experiments showed the same trends and/or significance in the results. We feel that the repeated statistical significance of our results despite this inherent variability speaks to the robustness of the observed effects of Pichia on inflammatory outcomes during AAD. The original data has been provided as a supplement to indicate this and the figure legends have been updated for clarity. The GATA3+ T cell measurement is the one exception to this as this antibody was only included in the flow cytometry panels in 2 experiments, and therefore Author response image 2 represents combined data from 2, rather than 3, experiments.

3. What was the rationale for adding IL-4+ cell analysis in Figure 2-supplement 1F but not in Figure 2? The authors mention that ICOS is associated with IL-4 production, but IL-4 data are not shown. Furthermore, In Figure 3-supplemental 2 include IL-5 and IL-13 to show that the same response characterized in Figure 2, but these metrics were not present in Figure 2. Please either include the data or explain.

Thank you for the opportunity to explain these discrepancies and we apologize for not making this more clear in the methods. The IL-4 antibody used for flow cytometry did not successfully stain our cells in at least 2 of our 3 neonatal exposure experiments, so this data was not included. The flow cytometer we used for these initial experiments also did not accommodate the additional colours to be able to add IL-5 and IL-13 to these neonatal exposure experiments. These additional markers were added to our panel in the germ-free experiment once we upgraded our machine. To address this reviewer’s concern, we attempted to perform IHC for IL-4 in lung sections taken from the experiments where mice were neonatally exposed to *P. kudriavzevii* but unfortunately were unsuccessful given the age of the samples. We hope to be able to further address the role of ICOS+ T cells and IL-4 production in future work.

4. In multiple places in the text it is stated that Pichia is absent from the gut at different time points, however no data was shown to directly support this conclusion. Either add data or delete these statements.

Our apologies for not clearly describing our experimental methods. The text has been revised to indicate that colonization occurs only during the period when the pups are housed with dams (who received the Pichia treatment and are also colonized during the treatment period), until weaning at 19 days of age.

Line 114-133: “To establish a causal role for early life fungal dysbiosis in asthma etiology and validate previous findings in the ECUAVIDA cohort, we exposed mice to *P. kudriavzevii* during the neonatal period and then used the house dust mite (HDM) model of AAD to induce airway inflammation at six weeks of age (Figure 1A,B). Pups were exposed to either *P. kudriavzevii* suspended in phosphate buffered saline (PBS) or PBS alone by painting the abdomen and face of lactating dams with these respective solutions every second day for two weeks following birth.”

Line 154-166: “Colony counts at day 16 (Figure 2A) and 21 (no colonies present) of life revealed that although levels were highly variable, *P. kudriavzevii* colonized the guts of pups born to dams treated for two weeks with this yeast until at least two days after the final treatment, but was no longer present in the gut microbiota after they were weaned on day 19 of life. Thus, pups were only colonized during the period when they were co-housed with dams and littermates, indicating that persistent exposure is required to maintain colonization (Figure 2-supplement 1a).”

After weaning, pups are moved to new cages and housed with same-sex littermates. Colonization during the first 3 weeks was assessed by colony counts and by plating fecal samples (when it was possible to collect fecal samples) from mice at days 14, 16, 18, and 21 of life. These data have been added as Figure 2-supplement 1. We plated the intestines of a small number of mice on day 9 of life by sacrificing these pups and found by qualitative observation of colony growth that there was abundant colonization. qPCR for Pichia at day 21 and 28 of life further confirmed barely/no detectable levels of Pichia in fecal pellets (defined as Ct >30 using primers specific for P. kudriavzevii), respectively. See Excel file provided to reviewers.

qPCR was done using the Qiagen Quantinova SYBR kit according to the manufacturer’s instructions and the following primers (3) (which we identified to be specific for *P. kudriavzevii* at Ct’s below 30):

F: CTGGCCGAGCGAACTAGACT

R: TTCTTTTCCTCCGCTTATTG

~169bp product.

Each reaction contained:

-2µl of template DNA

-QuantiNova SYBR Green master mix (Qiagen)

-Rox reference dye (Qiagen)

-H2O

-forward primer (10µM)

-reverse primer (10µM)

10µl reaction volume

The cycling protocol was as follows:

- 95^o^C for 2min

-40x: [(95^o^C for 5s) + (60^o^C for 30s)]

a. Line 114 – no data shown for colony counts at 21 days, remove from text or add this missing figure.

We agree that this is an important part of our findings, particularly because it speaks to the sensitivity of the early life window for immune development. We have struggled, however, with how to properly show a lack of colonies. Please see the added Figure 2-supplement 1 where we attempt to demonstrate this, visually.

b. Line 116 – no data is shown to support that Pichia is gone after weaning. Without showing that Pichia did not persist after weaning then, it is possible that prolonged fungal colonization is responsible for the AAD phenotype observed. Address this inconsistency by adding in the relevant data or addressing in the text.

We recognize that a lack of colony counts might difficult to represent, we therefore also performed PCR on the inoculum used in the GF experiment and found no evidence for the presence of Pichia based on the gel and qPCR (see added Excel file).

c. Line 120-124 – no data shown to support that there is no Pichia in the gut at 4 weeks.

Please see the added Figure 2-supplement 1 where we attempt to demonstrate this, visually.

d. Line 131-132 – have not shown fungal colonization ends at a particular day.

In addition to the additional colony counting data obtained through the revision experiment described in point 2 above, we performed ITS-2 sequencing on fecal pellets collected at four and eight weeks of age from our original experiments, but found no sequences that were annotated as Pichia (or Isaatchenkia orientalis, which is what the UNITE database calls P. kudriavzevii). This is also now clarified in the text:

Line 154-166: “Colony counts at day 16 (Figure 2A) and 21 (no colonies present) of life revealed that although levels were highly variable, *P. kudriavzevii* colonized the guts of pups born to dams treated for two weeks with this yeast until at least two days after the final treatment, but was no longer present in the gut microbiota after they were weaned on day 19 of life. Thus, pups were only colonized during the period when they were co-housed with dams and littermates, indicating that persistent exposure is required to maintain colonization (Figure 2-supplement 1a).”

When we performed a formal fungal microbiota sequencing analysis using the ITS86(F) and ITS4(R) primers in a subset of samples, all of the resulting sequencing files were <1Mb in size, indicating that very few fungi were detected (according to the standards used by Integrated Microbiome Resources, where the sequencing was done, the sequencing actually failed). Once we processed the sequencing data, very few samples passed the quality filtering steps and the fungal sequencing was therefore not done for all experiments (n=8 from stool collected at 4 weeks of age from animals treated neonatally (n=4 per treatment condition), n=8 from adolescent mice (6 weeks; n=4/treatment condition treated in adolescence), n=10 from stool collected at 8 weeks of age (n=2 controls, n=8 Pichia; data not fully processed given that only 3 samples had >150kb of raw data) at experimental endpoint after the HDM asthma model).

Interestingly, we noted that the number of reads/data recovered from samples collected at four weeks of age was higher than the number of reads we obtained when analyzing samples collected in adolescence or at experimental endpoint, even in Pichia-treated animals. After sequence quality filtering and taxonomic assignment, only 2 samples per treatment condition in the adolescent mice contained >500 fungal reads. All 8 samples from 4-week-old mice had at least 500 reads, but the fungal communities did not differ significantly in α or β (Bray-Curtis) diversity.

**Table resptable1:** 

Age	Treatment	Number of reads (after processing)
Adolescent (6 weeks)	Control	728
Adolescent (6 weeks)	Control	379
Adolescent (6 weeks)	Control	2038
Adolescent (6 weeks)	Control	308
Adolescent (6 weeks)	Pichia	552
Adolescent (6 weeks)	Pichia	466
Adolescent (6 weeks)	Pichia	143
Adolescent (6 weeks)	Pichia	873
4 weeks	Control	752
4 weeks	Control	750
4 weeks	Control	7485
4 weeks	Control	2128
4 weeks	Pichia	1922
4 weeks	Pichia	532
4 weeks	Pichia	842
4 weeks	Pichia	1220

Samples collected at 4 weeks of age from animals treated neonatally:

**Author response image 3. sa2fig3:** 

adonis(formula = braydist_ITSpn1 ~ Treatment, permutations = 4999)Permutation: free

Number of permutations: 4999

Terms added sequentially (first to last)

Df SumsOfSqs MeanSqs F.Model R2 Pr(>F)

Treatment 1 0.28239 0.28239 1.158 0.16178 0.2592

Residuals 6 1.46316 0.24386 0.83822

Total 7 1.74555 1.00000

**Author response image 4. sa2fig4:** 

Looking at the fungal genera identified, the majority of the fungi present were likely food-derived (P1 at the start of the SampleID indicates these mice were treated postnatally with Pichia or vehicle; the following C indicates control animals and P indicates Pichia animals):

**Author response image 5. sa2fig5:** 

Text has been added to indicate this:Line 170-174: “The absence of robust fungal communities in these animals at four and eight weeks of age was verified by assessing for the presence of fungi in DNA isolated from fecal samples using high-throughput sequencing and primers targeting the internal transcribed spacer region (ITS-2) of the fungal 18S rRNA gene (sequencing files generated <1Mb of raw data per sample).”

5. A discussion of data in Figure 3C-F correlates with previously published studies on bacterial dysbiosis and AAD would be a beneficial addition. Clostridales are mentioned as a direct contributor to SCFA, is the abundance of members of this genus altered in the experiment shown?

This is an excellent question, and one of particular interest for our lab. Please see our detailed response to point 9 below.

6. In Figure 3-supplemental 1, data is shown to support the hypothesis that the bacterial composition change in the gut alone does not account for the changes in immune response however no data is shown to verify that the fecal transplant procedure successfully recapitulated the Pichia exposed gut microbiome. An analysis comparing data in Figure 3D and Figure 3 supplemental 1B-D is important to show that the bacterial species present in the fecal sample obtained from the mice exposed to Pichia were able to successfully colonize the germ-free mice and be representative of the original population.

Thank you for this suggestion. This reviewer is correct in that colonization of germfree mice does not always reflect the donor. We were, therefore, pleased to see that all but one bacterial genus (Butyricicoccus-present at low abundance) transferred to the germ-free mice after removing low-frequency features and data filtering/rarefaction. Germ-free mice also had an expansion in the relative abundance of Akkermansia, Bifidobacterium, and Allobaculum, which was accompanied by a reduced relative abundance of *Lactobacillus* relative to the four-week-old SPF mice. This comparison has now been included in Supplementary file 2, and we feel it supports the important findings from this experiment.

It would also be interesting to do the same analysis with the post-AAD samples to show how Pichia exposed and germ-free colonized mice (Figure 3F and Figure 3-supplemental 1E-F) respond to AAD model – is there a difference?

We agree that this was an interesting analysis. While we feel it is outside of the scope of the present paper in the short report format, we have provided the results in Author response image 6 and Author response image 7.

Author response image 6 is the relative abundance of the genera identified in the germ-free animals at sacrifice:

**Author response image 6. sa2fig6:** 

Author response image 7 shows the genera identified in the fecal samples from the SPF mice at sacrifice:

**Author response image 7. sa2fig7:** 

Changes to the text:1. This study looks at the effects of Pichia on bacterial dysbiosis and immune response but does not show data to support a more general change in fungal dysbiosis (no other fungal species present in the mouse gut were assessed). Throughout the paper, as in line 92-95, this point should not be made without support.

Fungal dysbiosis in this study is defined as an outgrowth of *P. kudriavzevii* specifically. While transient, we still feel that this fits the definition of a change to the healthy-state microbial community. See response to point 4d. We have also clarified this in the text:

Line 110-113: “Accordingly, using overgrowth of *P. kudriavzevii* as a model of fungal dysbiosis, we sought to determine whether fungal dysbiosis in the neonatal period influences asthma outcomes later in life, and to identify which aspects of asthmatic immunopathology are affected.”

2. The biological relevance of Figure 2 and Figure 2-supplement 1 are difficult to interpret without first having established colonization/dysbiosis of the gut microbiome. Consider moving Figure 3 first to establish you can recapitulate the dysbiosis seen in children in this mouse model and then show in Figure 2 that this correlates with differences in the immune system later in life.

Thank you for this suggestion. We have now added a line to indicate that pups are colonized during the treatment period:

Line 133-135: “The presence of *P. kudriavzevii* in the guts of pups born to P. kudriavzevii-treated animals during the two-week treatment period was confirmed by colony counts from plated colon tissues (Figure 2-supplement 1a).”

We have also added a transition sentence to the next paragraph:

Line 151-154: “To further characterize fungal colonization in our model, we plated colon contents or fecal samples from pups born to *P. kudriavzevii*-treated dams immediately before and after weaning, when the gut microbiota is known to undergo dramatic shifts in community composition.”

3. The data in Figure 4 don't align with the other data in the paper without more discussion of the connection. Expand upon the discussion/relevance of assessing SCFA effects on Pichia in this paper. Have SCFA been shown to impact fungal colonization previously, in analyses that corelate SCFA with decreased AAD? Is the hypothesis being tested that before increased Pichia in children there was changes in the microbiome that decreased local SCFA concentrations and this allowed Pichia overgrowth, which in turn further impact the bacterial gut microbiome and immune development? Clarify a connection in the text as to why Figure 4 is important to the story arc of this paper. It might be useful to present both models in a concluding figure.

This is an excellent point and we are grateful to the reviewer for pointing this out. Clarity on the link between Pichia and the SCFA experiments is now provided by directly referencing the association between low levels of acetate and increased Pichia in the fecal samples from children in Ecuador. The references cited also indicate previous examples (especially from Gary Huffnagle’s group) showing that SCFAs affect the growth of fungi linked to AAD following antibiotic treatment. It has not previously been shown directly that SCFAs impact colonization. The rationale behind the SCFA investigation is that the neonatal gut of infants at risk of asthma is uniquely susceptible to transient colonization by microbes encountered by chance due to bacterial dysbiosis and a lack of SCFAs, and that transient colonization subsequently has the potential to alter both local gut microbiota communities and normal immune development. It is also possible that fungal colonization is a chance event that occurs in the general context of the neonatal gut, where colonization resistance is low, which then precipitates both bacterial and fungal dysbiosis during a critical window of concurrent immune development that increases an infant’s risk of developing asthma (i.e. we cannot say whether the bacterial or fungal dysbiosis occurs first). A detailed analysis/comparison of Clostridiales or SCFAs in the mouse samples collected after weaning would therefore not make biological sense (we suggest that reduced SCFA-producing bacterial communities precede fungal dysbiosis, rather than cause these changes, and Figure 2 shows bacterial communities after fungal dysbiosis has already occurred). We have made the following changes to add clarity:

Lines 231-236: “Given that SCFAs have also been demonstrated to protect against asthma development and to be reduced in abundance in stool from infants at risk of asthma in Ecuador (in conjunction with fungal dysbiosis), the CHILD cohort, and other birth cohorts, we next determined whether part of the asthma-protective effects of SCFAs could be mediated by preventing colonization of the infant gut by asthma-associated fungi.”

We have also added further clarification to situate these results within the larger context of the story:

Lines 289-294: “Taken together, our results suggest that gut bacterial communities with a reduced capacity for SCFA production in the guts of neonates at risk of asthma are permissive to invasion by transient fungal colonizers. Transient fungal colonization, in turn, may either directly or indirectly through disruption to the normal temporal succession of neonatal gut microbiota communities required for appropriate immune development, further alter immune development and susceptibility to asthma (Figure 4).”

We hope that our study will serve to inform future studies aimed at validating this work further, including correlation analyses between SCFA levels and fungal load in human data sets.

Thank you for this suggestion for an additional figure, a model putting the story together has now been included as Figure 4.

4. For Figure 3B, to address the relevance of circulating Pichia-specific IgG, compare data to fold changes in other studies in other systems.

Similar differences in IgG levels have been described by others:

Tropini et al. (2018) Cell. (4)

Castro-Dopico et al. (2019) Immunity.(5)

Doron et al. (2021) Cell. (6)

We agree that the response is subtle, but given that the measurement was done 1 week after weaning, when the mice are no longer colonized with Pichia, this is not necessarily surprising. When viewed in a format similar to other papers:

**Author response image 8. sa2fig8:** 

We have also performed an additional RNA-seq analysis to assess for differences in inflammatory markers/immune responses to Pichia in the guts of 16-day-old mice. These data support the hypothesis that early life colonization with *P. kudriavzevii* induces changes in the expression of immune-related genes (especially those related to dendritic cell function), including a downregulation of chymotrypsin-like genes (7,8), in the gut. These results have been added to Figure 2-supplement 1b.

5. Line 84 suggest that the data in Figure 1B are significant (p<0.05), but according to the figure they are not, edit text to acknowledge the p-value accordingly.

Thank you for pointing this out, we have removed this from the text.

Reviewer #2:In this report Boutin and colleagues have investigated the impact of exposing mice to P. kudriavzevii on HDM-induced allergic lung inflammation. The premise of the work comes from previous data from the CHILD study, amongst others, that link certain fungal species with an increased asthma severity.

Thank you for taking the time to thoroughly go through and evaluate our work, we appreciate your feedback.

There are two main conclusions from this work. First, the exposure must occur pre-weening and second, that SCFA directly reduce fungal adherence to intestinal epithelial cells.Figure 1 is in line with prior reports, setting the rationale for the author's focus on Pichia. The best control to include would be a comparison with another fungi, which might serve as a negative control. As the authors indicate, there are already a number of reports showing anti-fungal treatment, or colonisation with specific fungi, exacerbate allergic inflammation in mouse models. Is there something unique to Pichia, or is a general outgrowth of any fungi seen as increasing susceptibility?

This is a great question and one that is of ongoing interest to us. We have a separate manuscript under review describing a broader view of fungal communities in the guts of children from the CHILD cohort and have therefore refrained from including additional data on this in the present work. Based on our data, it would be an overgeneralization to say that it is a general outgrowth of fungi, however, it is unlikely that this is a phenomenon unique to Pichia. For example, there may be certain features conserved across groups of fungi (ex. the ability to form pseudohyphae or true hyphae (16,17)) that may make these fungi more likely to be associated with asthma outcomes. We attempted to touch on this in our discussion (line 269-285), but are happy to elaborate more if the reviewer would like.

The new data starts with Figure 2 where in general there is an increase in adaptive immune parameters in the Pichia group. Variability is very high, and in many cases, there appear to be 'responders' and 'non-responders', e.g. panel's D, E, G, H. Could these be cage affects? Statistical significance is reached in most cases, likely due to the pooling of experiments to have sufficient numbers. In this respect, it is unclear why data is pooled from 2 experiments in some case, and 3 in others? What is the basis for this? Overall, while there are some differences, particularly in the FACS analysis, the overall phenomenon does not look particularly robust, and disease (as opposed to FACS) parameters are largely missing, e.g. mucus production and lung function. There is no correlation with the level of Pichia colonisation per mouse and the readouts e.g. did the mice showing high levels of eosinophils also have higher CFU of Pichia during the 2 weeks of exposure?

This is an excellent point and one that has led to much discussion within our group. We similarly observed that there was a significant amount of variability in our data and have therefore examined this in detail. While these figures reflect pooled data, our three independent experiments showed the same trends and/or significance in the results. We feel that the repeated statistical significance of our results despite this inherent variability speaks to the robustness of the observed effects of Pichia on inflammatory outcomes during AAD. The original data has been provided as a supplement to indicate this and the figure legends have been updated for clarity. The GATA3+ T cell measurement is the one exception to this as this antibody was only included in the flow cytometry panels in 2 experiments, and therefore this figure represents combined data from 2, rather than 3, experiments.

With regards to the “responder” vs “non-responder” mice, we did observe that female mice appear to be more sensitive to the effects of fungal dysbiosis on AAD (note that female and male mice were housed separately and both sexes were included in the study (1 cage/sex that received Pichia/experimental repeat except for 1 of the 3 repeats)). This is likely in part due to the increased sensitivity of female mice to animal models of allergic disease.

Raising another important point, while ideally the correlation analysis between fungal load and asthma outcomes could be done, we found that this was not possible given the inability to (a) obtain fecal colony counts from pre-weaned neonates or (b) tightly regulate how much Pichia each pup received. Pups are only colonized with Pichia when they are housed with the dams (who are the ones that actually received the treatment), but lose colonization once they are weaned at 19 days of age. Therefore, we were unable to regulate exactly how much Pichia each pup received. Furthermore, before weaning, pups to not produce fully formed fecal samples so colonization has to be measured as an endpoint analysis-i.e. by collecting gut tissues. As a result, the colonization data is collected on different animals from those taken out to the AAD endpoint.

Furthermore, we agree with this reviewer that including additional data on disease parameters in the AAD would strengthen our conclusions. To address this reviewer’s concerns, we attempted to perform IHC for IL-4 in lung sections taken from the experiments where mice were neonatally exposed to *P. kudriavzevii* but unfortunately were unsuccessful given the age of the samples. We were, however, able to complement our findings using non-FACS-based assessments of airway inflammation by performing additional histology to assess for mucus production using PAS staining using similar methods as published by others (16). These results agreed with our other findings:

**Author response image 9. sa2fig9:** Goblet cell numbers quantified in primary (A), secondary (B), and tertiary (C) airways of mice in Figure 1. (D) Shows the average goblet cell index for all airways in each animal. Figure illustrates combined results of three experiments.

Additional histology methods: Lung sections from mice taken to end-point in the asthma model were stained with periodic acid-Schiff (PAS). Histological scoring was performed by a trained pathologist who was blinded to the study design and specimen IDs. One primary, one secondary, and one tertiary airway was selected from each slide and one hundred sequential airway epithelial cells were identified in each airway. The number of PAS+ cells (goblet cells) per hundred cells was counted and divided by the total number of epithelial cells to render a goblet cell index for each airway of each specimen. The average goblet cell index per section was obtained by averaging the goblet cell index of the three examined airways. When an airway was not visible in a section, this specimen was eliminated from the analysis. All slides were scored blinded.

Finally, we repeated the full 12-week early life exposure experiment and performed an analysis of cells in bronchioalveolar (BAL) washings following asthma induction. These results corroborated our FACS findings.

**Author response image 10. sa2fig10:** 

The concerns continue with Figure 2-sup where the authors have changed the protocol, now giving the Pichia per oral and post-weening. The conclusion is that the phenomenon no longer exists and thus the exposure must occur pre-weening. In fact, in this case there are far fewer animals and the figure legend indicates this is a single experiment (no repeats referred to?). The pathology score appears to have a similar increase in the Pichia group as in Figure 2, however an outlier in the control group likely stops significance. The fact that there is a lot of variability and so few mice, plus the fact that two different protocols have been used (skin administration vs oral gavage), means that the authors can't conclude the Pichia is only active in early life. These data do not appear robust and the conclusions are overstated.

Thank you for the feedback, these are important points and we appreciate the opportunity to clarify our methods and conclusions. We now realize that the rationale for including these results and the message we attempted to portray with these results was not clearly contextualized. We actually agree with this reviewer on many points and have therefore included this figure as a supplement rather than a main figure. Given the “critical window” hypothesis that bacterial dysbiosis in early life is critical to modulating atopy-associated outcomes later in life, we performed this experiment as a preliminary assessment of whether this hypothesis might hold true in the case of fungal dysbiosis as well. We recognized that it would be difficult to exactly replicate the methods from the early life experiments in older mice, but felt that using the technique of oral gavage would allow us to more tightly regulate how much *P. kudriavzevii* each animal received while maintaining exposure through the oral route. Given that we obtained negative results with this experiment and that the experiment was only repeated once, we have qualified our language to avoid overstating the results. It is still possible that, if the experiment were to be repeated, adolescent exposure would affect asthma outcomes, but our data do indicate that the effect is more pronounced with neonatal exposure. We hope the text reflects this nuance now:

Line 146-150: “Notably, mice exposed to *P. kudriavzevii* for two weeks in adolescent life (4-6 weeks of age) via oral gavage did not show evidence of increased lung inflammation in the context of HDM-induced AAD (Figure 1-supplement 1), highlighting the importance of the previously reported “critical window” of life during which the gut microbiota has the greatest ability to affect immune development relevant to asthma.”

In Figure 3 the highly variable exposure to Pichia is noted and the authors argue that there is an increase in Pichia specific IgG, however these data are shown as relative to control and there is a minor shift to around 1.2 in most samples. This analysis does not appear to be robust enough to conclude an adaptive response has been mounted against the Pichia.

Thank you for pointing out this important point. We have now performed an additional RNA-seq analysis in the guts of 16-day old mice to assess for differences in inflammatory markers/immune responses to Pichia. These data support the hypothesis that early life colonization with *P. kudriavzevii* induces changes in the expression of immune-related genes (especially those related to dendritic cell function), including a downregulation of chymotrypsin-like genes (7,8), in the gut. These results have been added to Figure 2-supplement 1b.

We also agree with this reviewer that our IgG data are subtle. However, given that the measurement was done 1 week after weaning, when the mice are no longer colonized with Pichia, we felt that these results are in line with existing literature. Indeed, similar differences in IgG levels have been described by others:

Tropini et al. (2018) Cell. (4)

Castro-Dopico et al. (2019) Immunity.(5)

Doron et al. (2021) Cell. (6)

The clearest data from Figure 3 comes with the GF recolonisation experiment, which in fact shows that transferring the microbiome from Pichia mice does not transfer any susceptibility. The inferred conclusion therefore is that the effect must be linked with early life immune maturation. However, no evidence is provided to support this. Neonatal GF mice would need to be colonised as a control, and immune parameters measured to prove that there has been some early life imprinting. Moreover, given the need to pool multiple experiments to show their phenomenon in Figure 2, similar numbers of mice are likely required throughout the manuscript to allow robust conclusions.

Thank you for this thoughtful feedback, this reviewer raises several points related to our experimental methods that have generated much discussion within our group. Our rationale for using germ-free mice in this model was based on previous work using germ-free mice as models of neonatal mice with a normal microbiota given the immunological immaturity of germ-free mice (see recent paper: (18)). Colonizing neonatal GF mice with our experimental set-up without also transferring Pichia to the GF mice would have been very challenging. Moreover, it has been shown that when fungi alone are provided to germ-free mice, AAD is actually improved (19), indicating that any bacteria/microbial stimulation can trigger the development of the immune system in these animals, and that the presence of bacteria is needed to add biological relevance to the experimental model. We therefore selected to perform the fecal transplant at the earliest time point we could be certain that *P. kudriavzevii* was not longer present, but bacterial changes in the microbiota could be identified. We have included Figure 2-supplement 2 to show that the bacterial microbiota communities separate according to the treatment condition of the original donor mice, as expected.

Finally, in Figure 4 the authors show that culturing the fungi in vitro in the presence of SCFA reduces fungal adherence to an intestinal epithelial cell line. The data is clear, but the obvious concern is the physiological relevance of this, and also the absence of any link with the prior data in the manuscript. Ideally the authors would include a better negative control, showing that other common intestinal metabolites do not affect fungal adherence. In addition, the SCFA should be utilised at ratios similar to that seen in vivo. In Figure 4, the highest concentration of 150umol/ml is utilised for each SCFA, however if the butyrate and propionate levels were relative to acetate, the conclusion would likely change. Direct cell toxicity of the SCFA should also be reported. These data would also carry more weight if in vivo experiments were performed to test whether there is reduced Pichia colonisation in early life in mice with higher SCFA levels.

These are all excellent points and we apologize for not taking the time in the paper to clearly outline our rationale and highlight the biological relevance of our methods and findings. To address these concerns, we used NaCl as a control for the adherence assay, and have performed an additional experiment using biologically relevant concentrations of SCFAs and biotin (an abundant factor produced by gut bacteria) as an additional control. Figure 3K has been updated to reflect the new results. When repeating this experiment, we also used biologically relevant molar ratios of the SCFAs.

We would like to clarify that SCFAs are no longer present in the media when the fungal cells are added to the colon cells (the Pichia cells are spun down and resuspended in a 50/50 solution of YPD and DMEM), so toxicity should not be a concern. Fungal cell stocks used to inoculate the TC7 cells were further plated to confirm viability and confirm that OD measurements tracked with cell counts (included in raw data for the new experiment). We also performed the growth curve and SEM experiments using molar ratios equivalent to concentrations found in the gut and found that butyrate and propionate had similar effects, even at the lower concentrations (see Author response image 11; note that previous studies have used similar concentrations of 100-150μmol/ml of butyrate as “biologically relevant” concentrations (17,20,21)).

We initially performed the epithelial cell adhesion assay as a proof-of-concept assay to demonstrate that reduced hyphae formation has a functional consequence for adherence. To further support these findings, we have performed an additional in vivo experiment to assess the ability of orally delivered biologically relevant molar ratios of the SCFAs to inhibit the colonization of the murine gut by P. kudriavzevii. We observed that mice supplemented with a cocktail of SCFAs in their drinking water exhibit a trend toward reduced colonization with *P. kudriavzevii* following antibiotic treatment and fungal oral gavage. This has been added as a new Figure 3-supplement 1. We cannot control the amount of Pichia each pup receives, so while we have attempted the early-life supplementation with SCFAs, it is difficult to determine how SCFAs impact colonization. For this reason, the in vivo experiments have been done in adult mice pre-treated with antibiotics.

Overall, although conceptually the manuscript is interesting, there are major concerns related to the robustness of the datasets and comparisons that are drawn between the different experimental systems.

Thank you for your feedback, we hope that we have clarified some of the methods and comparisons now.

**Author response image 11. sa2fig11:** (A-C) Growth over time (top) and optical density (OD) at 600nm at 18 hours (bottom) of *Pichia kudriavzevii* grown in Yeast Peptone Dextrose (YPD) broth supplemented with the sodium salts of the short chain fatty acids (SCFA) acetate (A), butyrate (B), or propionate (C) at the indicated concentrations. (D-G) Scanning electron microscopy of *P. kudriavzevii* grown in YPD (D) or the sodium salts of acetate (E), butyrate (F), or propionate (G). (A-C) Data represent results from three independent experiments performed in triplicate. Dots represent biological replicates and data are presented as mean ± SEM. Statistical comparisons are relative to SCFA-free controls. *p < 0.05, **p < 0.01, ***p < 0.001.

Reviewer #3:Boutin et al. performed a throughout study to investigate the cause of allergic airway disease induced by early fungal colonization. They used the commensal fungus Pichia kudriavzevii as reasonable example in mouse models. They furthermore demonstrate that the disease is only induced when mice are exposed early in life and that the resulting disease is predominantly induced by the fungus and not the resulting microbiome.I believe this is a well thought out and conducted study. The following points should be considered by the authors for improving the quality and clarity of their study:

Thank you for your positive feedback, we are grateful to this reviewer for taking the time to review our manuscript and for the opportunity to further clarify and expand upon our findings.

The major problem in the manuscript is the unjustified experiments with SCFAs. Suddenly in the manuscript there is a statement that SCFAs have been reported in the literature to inhibit fungi so the authors decided to test some of them for their inhibitory role on P. kudriavzevii. Subsequently and after the successful testing of the inhibitory role of the tested SCFAs in vitro the authors claimed that they have found the mechanism behind colonization of P. kudriavzevii. The authors could support this in several ways:- There is a huge variation in the P. kudriavzevii levels of the 115 subjects from the Child cohort. A 16S rRNA analysis of these subjects and a correlation analysis with P. kudriavzevii levels could prove that SCFA bacterial producers correlate negatively with P. kudriavzevii levels. The authors could measure the SCFAs in the stool samples of these children to confirm their hypothesis.

Thank you for highlighting this important point. In response to this feedback, we have taken the time to further clarify and more clearly outline the rationale for the SCFA experiments and link these findings to the rest of the paper/existing literature. While we have not directly measured SCFA levels in human stool samples used in the present analysis, our lab’s previous publication in Science Translational Medicine (Arrieta et al., 2015) measured fecal SCFA levels in stool samples from a subset of children in the CHILD cohort and found that acetate was reduced in samples from infants who developed atopy and wheeze later in life. Similarly, we found that acetate levels were reduced in samples collected at 3 months of age from infants in Ecuador who developed atopy and wheeze at age five years (Arrieta et al., 2018). These same infants also had increased Pichia in the same stool samples. This information has been clarified now:

Lines 231-234: “Given that SCFAs have also been demonstrated to protect against asthma development and to be reduced in abundance in stool from infants at risk of asthma in Ecuador (in conjunction with fungal dysbiosis), the CHILD cohort, and other birth cohorts(1,22–26).”

We also agree with this reviewer that our study does not identify the mechanism behind colonization, but rather generates several hypothsesis that require testing in future work. Clarity on the link between Pichia and the SCFA experiments is now provided by directly referencing the association between low levels of acetate and increased Pichia in the fecal samples from children in Ecuador. The rationale behind the SCFA investigation is that the neonatal gut of infants at risk of asthma is uniquely susceptible to transient colonization by microbes encountered by chance due to bacterial dysbiosis and a lack of SCFAs, and that transient colonization subsequently has the potential to alter both local gut microbiota communities and normal immune development. It is also possible that fungal colonization is a chance event that occurs in the general context of the neonatal gut, where colonization resistance is low, which then precipitates both bacterial and fungal dysbiosis during a critical window of concurrent immune development that increases an infant’s risk of developing asthma (i.e. we cannot say whether the bacterial or fungal dysbiosis occurs first). We have made the following changes to add clarity:

Lines 231-236: “Given that SCFAs have also been demonstrated to protect against asthma development and to be reduced in abundance in stool from infants at risk of asthma in Ecuador (in conjunction with fungal dysbiosis), the CHILD cohort, and other birth cohorts, we next determined whether part of the asthma-protective effects of SCFAs could be mediated by preventing colonization of the infant gut by asthma-associated fungi.”

We have also added further clarification to situate these results within the larger context of the story:

Lines 289-294: “Taken together, our results suggest that gut bacterial communities with a reduced capacity for SCFA production create conditions permissive to invasion by transient fungal colonizers. In the neonatal gut. Transient fungal colonization, in turn, may either directly or indirectly through disruption to the normal temporal succession of neonatal gut microbiota communities required for appropriate immune development, further alter immune development and susceptibility to asthma (Figure 4).”

We hope that our study will serve to inform future studies aimed at validating this work further, including correlation analyses between SCFA levels and fungal load in human data sets. To further bring together the 2 parts of our paper, a model putting the story together has now been included as Figure 4.

- In the mice experiments I would like to see that supplementation of the food with these SCFAs indeed leads to colonisation resistance against P. kudriavzevii.- Alternatively, a concomitant oral gavage of a known strong SCFA bacterial producer and P. kudriavzevii could offer the evidence needed to support the authors hypothesis.

Thank you for this excellent suggestion; this is an important point that our group has discussed at length in an attempt to identify the most appropriate experimental approach. We appreciate the opportunity to further elaborate on several experiments we have performed to address these questions.

We have performed an experiment where adult animals pre-treated with antibiotics (to allow for fungal colonization and mimic the neonatal gut) were supplemented with short-chain fatty acids and orally gavaged with P. kudriavzevii. Specifically, six to seven-week-old male and female mice were housed two per cage and drinking water was supplemented with 0.5mg/mL cefoperazone (Σ catalog #62893-20-3) as previously described (27) on Days 0-3 to clear the intestinal bacterial microbiota. Half of the mice further had their water supplemented with a cocktail of SCFAs according to previously established protocols (23,28) for the duration of the experiment. The cocktail consisted of sodium acetate (67.5mM), sodium propionate (25.9mM), and sodium butyrate (40mM), and the control animals received water that was pH and sodium matched (28). All water was filter sterilized and had Splenda added (8g/L) to improve palatability. On Day 3, all mice were given an oral gavage with 10^7^ cells of *P. kudriavzevii* obtained from a 48-hour culture generated from a single colony of yeast and grown at 37°C while shaking. Two days after the gavage, fecal samples were collected for plating. Uncolonized mice were removed from the analysis.

We observed that mice supplemented with a cocktail of SCFAs in their drinking water exhibit a trend toward reduced colonization with *P. kudriavzevii* following antibiotic treatment and fungal oral gavage. This has been added as a new Figure 3-supplement 1. We acknowledge that this experiment has limitations as a result of differences in water consumption by the mice and other technical challenges. Due to experimental restrictions resulting from the Covid-19 pandemic, however, we were unable to troubleshoot the experimental protocol but feel that the results still support our conclusions. We feel that our epithelial cell adhesion assay is a realistic representation of the influence of SCFAs on fungal cell colonization and is sufficient to support our claims.

Lines 248-257: Furthermore, mice supplemented with a cocktail of SCFAs in their drinking water exhibited a trend toward reduced colonization with *P. kudriavzevii* following antibiotic treatment and fungal oral gavage (Figure 3-supplement 1).

Furthermore, we performed a pilot study wherein we supplemented dams with 4mg/mL Splenda and 20mM sodium acetate in the drinking water (intervention group; n=4 but one dam who received *P. kudriavzevii* perished due to dehydration) according to previously described methods (22) or water supplemented with Splenda only (control group; n=2) beginning at 14 days gestation. Water was changed every two days until the pups were weaned at three weeks of age. Half of the dams in each water treatment condition were treated with *P. kudriavzevii* as described in the methods section. The house dust mite (HDM) model of AAD was then used to induce asthma in the pups once they reached six weeks of age. In these pups, we observed a decreased in the severity of AAD in animals treated with acetate, regardless of *P. kudriavzevii* exposure status, making it difficult to draw direct conclusions of the ability of SCFAs to protect against AAD through a reduction in fungal colonization. These findings further supported our decision to use the epithelial cell adhesion assay rather than in vivo experiments to directly show the ability of SCFAs to affect the ability of fungal cells to adhere to epithelial cells in the gut.

**Author response image 12. sa2fig12:** Number of eosinophils (left) and percentage of IL-17+ (middle) and ICOS+ (right) T cells in the lungs of mice given allergic airway disease via house dust mite extract sensitization and challenge in adulthood. Gating strategies are indicated on the y-axis. Animals were neonatally exposed to the yeast *Pichia kudriavzevii* (Pichia) or PBC (control) and born to dams supplemented with either both acetate and 4mg/mL Splenda and 200mM sodium acetate (acetate) or Splenda alone (Splenda) in drinking water.

One of the most interesting findings was that: unlike the neonatal exposure, mice exposed to P. kudriavzevii for two weeks in adolescent life (4-6 weeks of age) via oral gavage did not demonstrate increased lung inflammation. However, it is not completely clear to me whether the two exposure groups ("neonatal exposure" and "adult exposure" as stated in the Methods) have the same treatment methods and dosage. Please elaborate more on this.

Thank you for your interest in this finding! We agree that this is one of the most interesting aspects of our analysis and appreciate the opportunity to elaborate. Given the “critical window” hypothesis that bacterial dysbiosis in early life is critical to modulating atopy-associated outcomes later in life, we performed this experiment as a preliminary assessment of whether this hypothesis might hold true in the case of fungal dysbiosis as well. We recognized that it would be difficult to exactly replicate the methods from the early life experiments in older mice, but felt that using the technique of oral gavage would allow us to more tightly regulate how much *P. kudriavzevii* each animal received while maintaining exposure through the oral route. While it was not possible to regulate exactly how much *P. kudriavzevii* each pup received in the neonatal experiments, we felt that the oral gavage technique would represent an improved method for the adolescent exposure experiments. We also used a 2-week treatment period to more closely replicate our neonatal experiment methods. Given that we obtained negative results with this experiment and that the experiment was only repeated once, we have qualified our language to avoid overstating the results. We hope the text reflects this nuance now:

Line 146-150: “Notably, mice exposed to *P. kudriavzevii* for two weeks in adolescent life (4-6 weeks of age) via oral gavage did not show evidence of increased lung inflammation in the context of HDM-induced AAD (Figure 1-supplement 1), highlighting the importance of the previously reported “critical window” of life during which the gut microbiota has the greatest ability to affect immune development relevant to asthma.”

The authors stated in Line 387 of the Methods section that "Successful transfer of colonic bacteria was verified via 16S sequencing", but I did not find relevant results or figures.

Thank you for pointing this out, we now realize that this was not sufficiently fleshed out in the text and have made some changes to clarify this point. We have included Figure 2-supplement 2 to show that the bacterial microbiota communities separate according to the treatment condition of the original donor mice, as expected. All but one bacterial genus (Butyricicoccus-present at low abundance) transferred to the germ-free mice after removing low-frequency features and data filtering/rarefaction.

Line 128: “Bacterial populations of P. kudriavzevii-exposed and -naïve mice separated moderately”Please supply an effect size measure in the corresponding figure or text to elucidate what a ‘moderate’ effect is in this context.

The term “moderately” has been removed for clarity and the p-value (p=0.01) has been added to the text.

Line 131: High variability and low sample size for a mice-based 16s rRNA analysis, in Figure 3C, 3D, 3F (as main results), although being statistically significant. The authors used Bray-Curtis Dissimilarity as community β-diversity measures, I wonder whether other distance measures (e.g. weighted/unweighted UniFrac) could have higher power in discriminating the two groups.

Thank you for your feedback. We have controlled for variance caused by cage effects (as described in the methods), which is why the R2 and p-values are not as dramatic as you might expect based on looking at the figure. We also saw that mice separated according to treatment condition in an independent experiment. We obtained very similar results using UniFrac metrics, and in some instances the effect size (R2) was less than when using the Bray-Curtis dissimilarity.

Line 132: I would suggest adding the points for individual samples in Figure 3E.

Done for 3E and Figure 2-supplement 2.

Line 272: Is four-week-old mice really "adult exposure" as stated in the subtitle? or, adolescent?

This is now changed to adolescent.

Line 383: were the recipient germ-free mice 12-16-week-old? Based on the figure legend of Figure 3-supplement 1a, AAD induction was at day 42 (week 6). Please confirm and/or clarify.

The recipient mice were 12-16 weeks old. The day 42 refers to the experimental timeline. This is now clarified in the figure legend:

(a) Experimental design (numbers indicate days of experimental timeline beginning from birth of donor mice).

Figure 1: As statement by the authors, the variation is high with respect to fungal counts. If possible, scatter or bee swarm plots would help to assess the variation better.

See response to Reviewer 1, point 1.

Figure 2c In the caption, the pathology score is shown and referenced as "left; 5-20". However, controls clearly show a value below 5. Please correct the range or elucidate.

Thank you for noticing this. This was a mistake, as the range is 4-20 (4 parameters scored on a scale of 1-5). The legends have been updated.

Figure 2-suppl c: Same as with Figure 2C, please correct the caption.

This has been corrected.

Figure 3c-d: Please report a proper effect size in the plot (such as F-values or R2) or in the caption in addition to the p-value.

R2 values have been added here and in the Figure 2-supplement 2.

Figure 3—figure supplement c "Relative abundances of the top 100 sequences at genus level […]""Top 100 sequences" does not make sense in this context. Please correct or elucidate. I guess it should be similar to Figure 3—figure supplement E.

Thank you. This has been changed to indicate that it includes all identified genera in the data sets.

Figure 3-suppl e "top 100 genera"Only 19 genera are shown in the plot (which is reasonable). Please report it properly in the caption.

Corrected.

Please lower case the sub-figure letters in the figure captions (or vice versa).

Thank you, this has been corrected.

References:

1. Arrieta MC, Arévalo A, Stiemsma L, Dimitriu P, Chico ME, Loor S et al. Associations between infant fungal and bacterial dysbiosis and childhood atopic wheeze in a nonindustrialized setting. J Allergy Clin Immunol 2018;142:424-434.e10.

2. Douglass AP, Offei B, Braun-Galleani S, Coughlan AY, Martos AAR, Ortiz-Merino RA et al. Population genomics shows no distinction between pathogenic Candida krusei and environmental Pichia kudriavzevii: One species, four names. PLoS Pathog 2018;14:e1007138.

3. Carvalho A, Costa-De-Oliveira S, Martins ML, Pina-Vaz C, Rodrigues AG, Ludovico P et al. Multiplex PCR identification of eight clinically relevant Candida species. Med Mycol 2007;45:619–627.

4. Tropini C, Moss EL, Merrill BD, Ng KM, Higginbottom SK, Casavant EP et al. Transient Osmotic Perturbation Causes Long-Term Alteration to the Gut Microbiota. Cell 2018;173:1742-1754.e17.

5. Castro-Dopico T, Dennison TW, Ferdinand JR, Mathews RJ, Fleming A, Clift D et al. Anti-commensal IgG Drives Intestinal Inflammation and Type 17 Immunity in Ulcerative Colitis. Immunity 2019;50:1099-1114.e10.

6. Doron I, Leonardi I, Li X V, Kusakabe T, Puel A, Iliev ID. Human gut mycobiota tune immunity via CARD9- dependent induction of anti-fungal IgG antibodies Article Human gut mycobiota tune immunity via CARD9-dependent induction of anti-fungal IgG antibodies. Cell Published Online First: 2021. doi:10.1016/j.cell.2021.01.016.

7. Naujokat C, Berges C, Höh A, Wieczorek H, Fuchs D, Ovens J et al. Proteasomal chymotrypsin-like peptidase activity is required for essential functions of human monocyte-derived dendritic cells. Immunology 2007;120:120–132.

8. Chiba S, Ikushima H, Ueki H, Yanai H, Kimura Y, Hangai S et al. Recognition of tumor cells by Dectin-1 orchestrates innate immune cells for anti-tumor responses. *eLife* 2014;3:1–20.

9. Boutin RCT., Sbihi H, Dsouza M, Malhotra R, Petersen C, Dai D et al. Mining the infant gut microbiota for therapeutic targets against atopic disease. Allergy 2020;published. doi:https://doi.org/10.1111/all.14244.

10. Arrieta MC, Stiemsma LT, Dimitriu PA, Thorson L, Russell S, Yurist-Doutsch S et al. Early infancy microbial and metabolic alterations affect risk of childhood asthma. Sci Transl Med 2015;7:307ra152.

11. Willart MAM, Deswarte K, Pouliot P, Braun H, Beyaert R, Lambrecht BN et al. Interleukin-1α controls allergic sensitization to inhaled house dust mite via the epithelial release of GM-CSF and IL-33. J Exp Med 2012;209:1505–1517.

12. Schuijs MJ, Willart MA, Vergote K, Gras D, Deswarte K, Ege MJ et al. Farm dust and endotoxin protect against allergy through A20 induction in lung epithelial cells. Science 2015;349:1106–1110.

13. Shao TY, Ang WXG, Jiang TT, Huang FS, Andersen H, Kinder JM et al. Commensal *Candida albicans* Positively Calibrates Systemic Th17 Immunological Responses. Cell Host Microbe 2019;25:404-417.e6.

14. Park J, Kim M, Kang SG, Jannasch AH, Cooper B, Patterson J et al. Short-chain fatty acids induce both effector and regulatory T cells by suppression of histone deacetylases and regulation of the mTOR–S6K pathway. Mucosal Immunol 2015;8:80–93.

15. Zhang Z, Biagini Myers JM, Brandt EB, Ryan PH, Lindsey M, Mintz-Cole RA et al. β-Glucan exacerbates allergic asthma independent of fungal sensitization and promotes steroid-resistant TH2/TH17 responses. J Allergy Clin Immunol 2017;139:54-65.e8.

16. Skalski JH, Limon JJ, Sharma P, Gargus MD, Nguyen C, Tang J et al. Expansion of commensal fungus Wallemia mellicola in the gastrointestinal mycobiota enhances the severity of allergic airway disease in mice. PLoS Pathog 2018;14:e1007260.

17. Noverr MC, Huffnagle GB. Regulation of *Candida albicans* Morphogenesis by Fatty Acid Metabolites. Infect 2004;72:6206–6210.

18. Bernardes EVT, Kucha V, Gutierrez MW, Laforest-lapointe I, Jendzjowsky NG, Cavin J et al. in microbiome assembly and immune. 2020;:1–16.

19. Jiang TT, Shao TY, Ang WXG, Kinder JM, Turner LH, Pham G et al. Commensal Fungi Recapitulate the Protective Benefits of Intestinal Bacteria. Cell Host Microbe 2017;22:809-816.e4.

20. Schulthess J, Pandey S, Capitani M, Rue-Albrecht KC, Arnold I, Franchini F et al. The Short Chain Fatty Acid Butyrate Imprints an Antimicrobial Program in Macrophages. Immunity 2019;50:432-445.e7.

21. Smith PM, Howitt MR, Panikov N, Michaud M, Gallini CA, Bohlooly-Y M et al. The microbial metabolites, short-chain fatty acids, regulate colonic Treg cell homeostasis. Science 2013;341:569–573.

22. Thorburn AN, McKenzie CI, Shen S, Stanley D, Macia L, Mason LJ et al. Evidence that asthma is a developmental origin disease influenced by maternal diet and bacterial metabolites. Nat Commun 2015;6:7320.

23. Cait A, Hughes MR, Antignano F, Cait J, Dimitriu PA, Maas KR et al. Microbiome-driven allergic lung inflammation is ameliorated by short-chain fatty acids. Mucosal Immunol 2018;11:785–795.

24. Trompette A, Gollwitzer ES, Yadava K, Sichelstiel AK, Sprenger N, Ngom-Bru C et al. Gut microbiota metabolism of dietary fiber influences allergic airway disease and hematopoiesis. Nat Med 2014;20:159–166.

25. Arrieta M, Stiemsma LT, Dimitriu PA, Thorson L, Russell S, Yurist-doutsch S et al. Early infancy microbial and metabolic alterations affect risk of childhood asthma. Sci Transl Med 2015;7:ra152.

26. Roduit C, Frei R, Ferstl R, Loeliger S, Westermann P, Rhyner C et al. High levels of butyrate and propionate in early life are associated with protection against atopy. Allergy Eur J Allergy Clin Immunol 2019;74:799–809.

27. Noverr MC, Falkowski NR, Mcdonald RA, Mckenzie AN, Huffnagle GB. Development of Allergic Airway Disease in Mice following Antibiotic Therapy and Fungal Microbiota Increase: Role of Host Genetics, Antigen, and Interleukin-13. Infect Immun 2005;73:30–38.

28. Smith PM, Howitt MR, Panikov N, Michaud M, Gallini CA, Bohlooly-Y M et al. The microbial metabolites, short-chain fatty acids, regulate colonic Treg cell homeostasis. Science 2013;341:569–573.